# Effect of connection induced upper body movements on embodiment towards a limb controlled by another during virtual co-embodiment

Harin Hapuarachchi[1]*, Takayoshi Hagiwara[2], Gowrishankar Ganesh[3], Michiteru Kitazaki[1]*

1 Department of Computer Science and Engineering, Toyohashi University of Technology, Toyohashi, Aichi, Japan, 2 Graduate School of Media Design, Keio University, Kanagawa, Japan, 3 Laboratoire d'Informatique de Robotique et de Microelectronique de Montpellier (LIRMM), Univ. Montpellier, CNRS, Rue Ada, Montpellier, France

* harinmanujaya@gmail.com (HH); mich@tut.jp (MK)

**Data Availability Statement:** All raw data files are available at Mendeley Data https://data.mendeley.com/datasets/3r53jcd7pc/1 doi:10.17632/3r53jcd7pc.1.

## Abstract

Even if we cannot control them, or when we receive no tactile or proprioceptive feedback from them, limbs attached to our bodies can still provide *indirect* proprioceptive and haptic stimulations to the body parts they are attached to simply due to the physical connections. In this study we investigated whether such indirect movement and haptic feedbacks from a limb contribute to a feeling of embodiment towards it. To investigate this issue, we developed a 'Joint Avatar' setup in which two individuals were given full control over the limbs in different sides (left and right) of an avatar during a reaching task. The backs of the two individuals were connected with a pair of solid braces through which they could exchange forces and match the upper body postures with one another. Coupled with the first-person view, this simulated an experience of the upper body being synchronously dragged by the partner-controlled virtual arm when it moved. We observed that this passive synchronized upper-body movement significantly reduced the feeling of the partner-controlled limb being owned or controlled by another. In summary, our results suggest that even in total absence of control, connection induced upper body movements synchronized with the visible limb movements can positively affect the sense of embodiment towards partner-controlled or autonomous limbs.

## Introduction

In daily life, we are usually aware of which objects/limbs we perceive belong to us and have certain perceptual and motor expectations from them. This is due to the sense of 'embodiment' we feel towards these limbs. The concept of sense of embodiment is phenomenologically complex and the term embodiment has been defined differently in various contexts. From a philosophical perspective, embodiment is considered to be how a person defines and experiences

**Funding:** This research was supported by the Japan Science and Technology Agency's ERATO grant (number JPMJER1701) (Inami JIZAI Body Project) and the Japan Society for the Promotion of Science's KAKENHI grant (number JP20H04489, JP22H04774). The funders had no role in study design, data collection and analysis, decision to publish, or preparation of the manuscript.

**Competing interests:** The authors have declared that no competing interests exist.

himself/herself [1, 2]. On the other hand, embodiment science in virtual reality (VR) research studies the effects of self-avatars on its users [3]. Embodied avatars are defined to be avatars that are co-located with the user's body and seen from a first-person perspective within an immersive virtual environment [4]. Here we are interested in sense of embodiment, as defined in cognitive neuroscience, in which embodiment is related to how the brain expresses the body and is defined to consist of three subcomponents [4]: the 'sense of self-location', the 'sense of agency' and the 'sense of ownership'. Sense of self-location is known as the ability to perceive the location of one's body parts [1, 5], sense of agency is the sense of having control of motion [1, 6], and sense of ownership refers to one's self-attribution of the body [7].

A variety of both VR and real-life studies have investigated these subcomponents of sense of embodiment and have shown that, given the right sensory and/or motor stimulations, humans can elicit embodiment of virtual and physical objects other than their own bodies [8, 9]. For example, previous studies have shown that humans can have illusory body ownership to a child or a small doll [10–12], slimmer and wider mannequin bodies [13], bodies in a different skin color [14], a longer arm [15], animal bodies [16], and even towards robots [17–22]. Furthermore, embodiment can also be elicited when different body parts are re-associated [23] or even when the body is fully or partially invisible [24–28].

Many of these studies have highlighted haptic feedback in synchrony with visual feedback as a promising mode of induction of sense of ownership [29–31]. With the development of VR and robotic devices, studies have shown the benefits and usage of haptic devices in virtual reality with regards to presence [32, 33], performance [33], and learning [34]. Recently some studies have shown that different types of haptic feedbacks such as force, and vibro-tactile feedback can affect embodiment differently [35, 36].

But almost all previous embodiment studies have considered scenarios where a real limb is replaced by an artificial limb (like a rubber hand). In these scenarios, tactile feedback [29, 31] or proprioceptive feedback [37–39] corresponding to visible touch or movement [38, 39] of the artificial limb, can be provided on the real limb it replaces. However, such scenarios are not possible with 'Independent additional limbs' [40], like in the case of robotics prosthetics or robotics supernumerary limbs, as there is no replacement limb in these scenarios to provide a movement or tactile feedback on.

Previous studies by Kalckert & Ehrsson [41, 42] have shown that passive synchronous movement of a participant's finger along with the corresponding finger of a rubber hand (while the participant observes only the rubber hand) induces an equally strong sense of ownership towards the rubber hand compared to the traditional rubber hand illusion which involves visuo-tactile stimulation with a paint brush. However, if we try to induce embodiment towards an artificial limb or a prosthetic limb of an amputee, the passive synchronous movements of the artificial limb (or the autonomous prosthetic limb) will not provide proprioceptive feedback of the whole limb movements to the amputee. Limb movements will only lead to forces and hence movements of the body since the artificial limb is physically connected to the user's body. It is unknown whether such connection induced body movements caused by the connection forces can increase the sense of ownership and/or agency towards partner-controlled limbs while the user/amputee observes its movements.

To investigate this issue, we developed a novel virtual joint avatar setup where two individuals could be immersed/co-embodied within the same avatar in first-person perspective while each controlled the limbs on one side of the body (left-side and right-side limbs). The joint avatar setup in this study included a novel back brace setup (Fig 2A), that connected the dyad partner's shoulders, ensuring that the changes in upper body posture due to the movement of an arm by one partner was transmitted to the other. Therefore, in our setup, the upper body of an individual moved not only when the participant made arm movements, but also when the

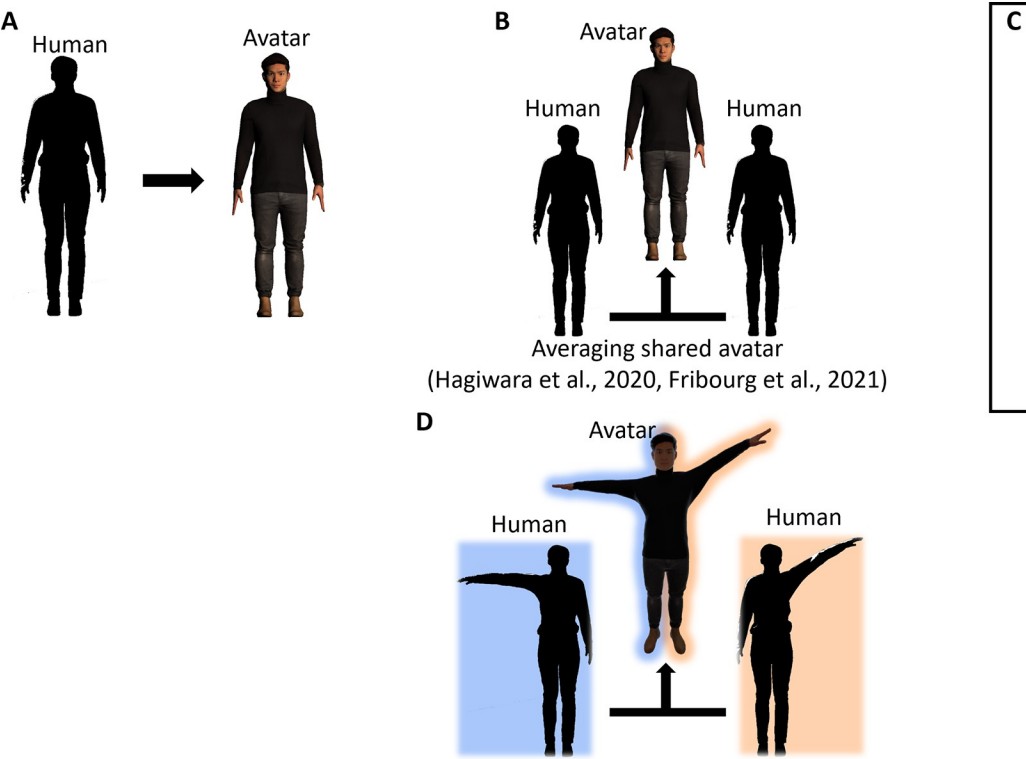
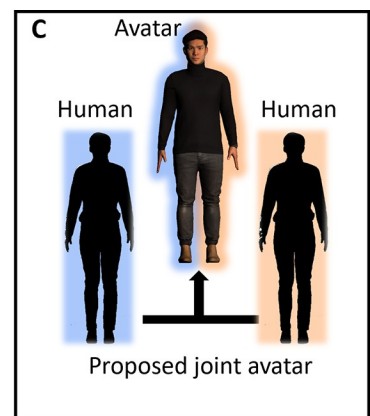

**Fig 1. Proposed avatar with left and right-side sharing between two people.** (A) An avatar is controlled by an individual conventionally. (B) A shared avatar or co-embodiment avatar is controlled by two individuals, and their movements are averaged and reflected in the shared avatar. (C) The proposed avatar is a joint avatar in which body parts are controlled by different individuals. (D) Example posture of our joint avatar, and the two individuals controlling it. The figures were created using unity2017.4.1f1 (https://unity3d.com/get-unity/download/archive).

individual saw the other arm being moved by participant's partner (and similar to how it would have moved if he/she had made arm movement himself/herself).

As shown in Fig 1A, in previous studies related to avatar embodiment in VR, conventionally one-to-one scenarios where one individual embodies one full avatar, or a robot, controlling its movements alone have been reported mostly. A few studies [43, 44] have examined "shared" avatars, whose movements were determined by averaging two individual's motions in real time (Fig 1B). However, there is no reported study that investigates embodiment, especially ownership towards limbs or parts of an avatar attached to a person, but completely controlled by another when it comes to VR or robotics experiments. That is, embodiment towards limbs fully controlled by a different person from which one does not get any direct tactile or proprioceptive information from the limb except the connection force feedbacks (forces due to the physical connection between the arm and the upper body) and the resulting passive upper body movements (Fig 1C). One study by Wegner et al. (2004) [45] on vicarious agency (that did not use virtual avatars or robotics) has reported proof about experiencing control over the movements of others with a setup where participants watched themselves in a mirror while another person behind them, hidden from view, extended hands forward on each side where participants' hands would normally appear and performed a series of movements. They have shown that hearing instructions previewing hand movements causes an enhanced feeling of controlling the hands [45]. However, factors other than verbal instructions that may affect control or ownership towards limbs controlled by others remain unexplored.

Using the proposed joint avatar in this study, we conducted an experiment with a two per-son (dyad) 'joint task' in VR. Dyads in our joint task stood with their backs to each other and received the first-person view from the eyes of the same 'joint avatar'. Each partner in the dyad controlled one arm of the joint avatar (Fig 1D; see S1 Movie). The right arm of the joint avatar copied the movements of the right arm of one dyad partner (tracked using optical marker positions read from a Vicon motion capture system), and the left arm of the joint avatar copied the left arm of the other dyad partner. The task presented to the dyads required them to reach targets presented to the left side (in front of them) with the right hand of the avatar, and con-versely, reach targets presented to the right side of the avatar, with the left hand of the avatar. All reaches thus required them to twist their upper bodies. The novel back brace setup shown in Fig 2A along with the pre-determined target positions for the reaching task were crucial to accurately convey/receive the upper body posture to/from the partner to the upper bodies of the participants in synchrony with the visual feedback of the joint avatar movements simulat-ing a feeling of the upper body being dragged along with the movements of the partner-con-trolled virtual arm. Since our experiment involved healthy participants, we immobilized their real arms, on the side where they experienced the virtual avatar arm moved by the partner, by tying it to their motion capture suits so that there would be close to no proprioception from that arm. Participants were asked to relax and not use their immobilized arm during the experiment.

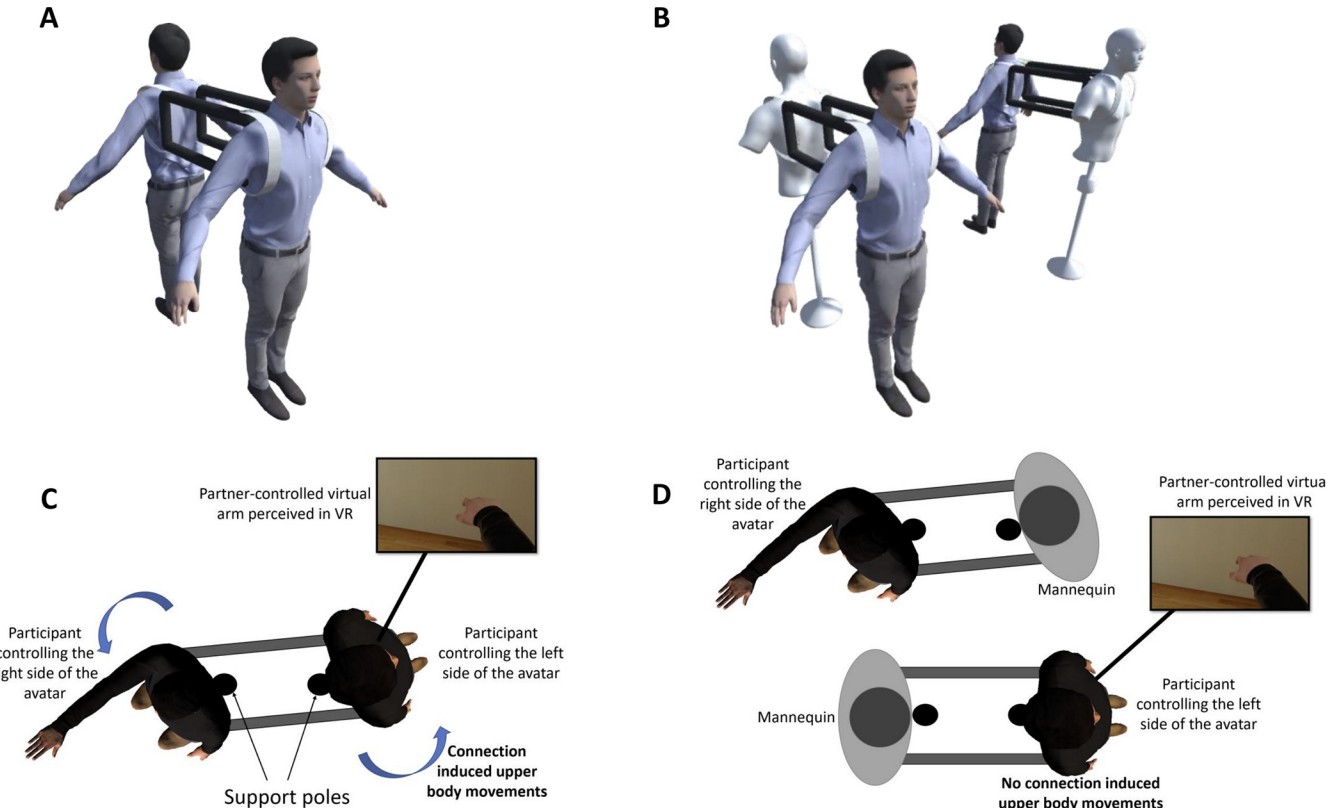

**Fig 2. Experiment setup.** The participants worked in dyads and stood back-to-back, resting against two 'support poles'. They wore the braces on their back similar to wearing a backpack. The brace consisted of four stiff horizontal rods set between their backs of the two bodies it connected. The participants worked in two conditions. **(A)** In the *Tied* condition, the backs of the partners were connected by the brace ensuring **(C)** that any body twists made by one participant during a reaching movement were transmitted to the other. **(B)** In the *Separated* condition, the shoulders of the participants were connected to a mannequin. **(D)** Thus, while the participants still saw the arm movements by their partner, they did not receive any passive upper body movements corresponding to the movements. The figures were created using blender2.93.4 (https://www.blender.org/download/lts/2-93/) and unity2017.4.1f1 (https://unity3d.com/get-unity/download/archive).

Each time the participant controlling the right side of the joint avatar moved his right arm to reach the targets, the back brace ensured that the real right shoulder of his partner was synchronously pushed to the front. Similarly, each time the participant controlling the left side of the joint avatar moved his left arm to the front to reach targets, the left shoulder of his partner was pushed to the front synchronously due to the back brace.

We evaluated the senses of ownership and agency perceived by the participants towards the arm they controlled (*controlled arm*) and towards the arm controlled by their partner (*non-controlled arm*) in two conditions. The *Tied* condition (Fig 2A and 2C), where they received connection induced upper body movements from the non-controlled arm, and the *Separated* condition (Fig 2B and 2D) in which each dyad participant was attached to a mannequin during the reaching task, again using the back brace.

Body representation refers to perception, memory, and cognition related to the body and is updated continuously by sensory input [46]. Ownership may be measured using changes in arousal levels using galvanic skin resistance. However, GSR measures are noisy especially in tasks like ours that involve movement. On the other hand, several studies have shown that embodiment and or ownership are often correlated with changes in body representations like proprioceptive drift [29, 31] and visual localization of the real body [1, 47]. Proprioceptive drift measures changes in the so called 'body schema', while the visual tests show changes in the 'body image' of the participant. Here motivated by these studies we conducted three behavioral tests to find and compare changes in the body representation after the experience of our joint avatar and corresponding connection induced upper body movements. Specifically, we purposely presented the non-controlled arm and shoulder displaced by 10 cm from a participant's real shoulder, expecting to find possible differences in changes in their perceived shoulder location and shoulder width between the Tied and Separated conditions.

Joint/shared/co-embodied avatars and robots can provide means for people with different physical disabilities to combine their strengths with others and work together in one complete avatar/robot to perform complicated tasks effectively while minimizing their physical disabilities at the same time. For example, a person missing a left arm can pair up with a person missing a right arm to co-embody one joint avatar and control different sides together as one whole avatar in VR. However, in such cases, since one half of the body is completely controlled by another, it becomes necessary to figure out ways to increase the sense of embodiment (induce illusory embodiment) to body parts controlled by the partner to minimize discomfort. It is difficult to imagine that two persons are physically connected in the real world. However, the force on the upper body could be conveyed to a person at a distant place with appropriate devices. It is much easier than conveying detailed proprioceptive senses of the whole limb. Thus, our experimental paradigm will contribute to developing joint avatars and robots in the future.

In summary, the purpose of this experiment was to simulate a feeling similar to the movements felt on the torso of an amputee wearing a prosthetic arm when the prosthetic arm moves by itself and to investigate if such movements influence the senses of ownership and agency felt towards the prosthetic arm. Therefore, we made the following two hypotheses regarding the Tied condition in which such upper body movements were present and the Separated condition in which such upper body moves were absent while the partner-controlled arm moved.

H1: Senses of ownership and agency towards the virtual non-controlled arm would be higher in the Tied condition compared to the Separated condition.

H2: Proprioceptive drifts towards the virtual non-controlled arm would be higher in the Tied condition compared to the Separated condition.

## Results

### Sense of body ownership towards the non-controlled arm became less negative in Tied condition over Separated condition

Questionnaire at the end of each session consisted of 8 questions, 4 regarding the left hand and 4 regarding the right hand. Participants answered all the questions (Q1. I felt as if the virtual left hand I saw was my left hand, Q2. It felt as if the virtual left hand I saw was someone else, Q3. It felt like I could control the virtual left hand as if it were my own left hand, Q4. I felt as if the virtual left hand was moving by itself, Q5. I felt as if the virtual right hand I saw was my right hand, Q6. It felt as if the virtual right hand I saw was someone else, Q7. It felt like I could control the virtual right hand as if it were my own right hand, Q8. I felt as if the virtual right hand was moving by itself). Questionnaire answers were conducted in a 7-point Likert scale from -3 to +3 where -3 meant "Strongly disagree" and +3 meant "Strongly agree".

Using the answers to the questionnaires, senses of ownership and agency towards left and right sides of the joint virtual avatar were calculated as follows referring to [48].

Sense of body ownership towards the left side  = Q1 –Q2
Sense of agency towards the left side               = Q3 –Q4
Sense of body ownership towards the right side = Q5 –Q6
Sense of agency towards the right side             = Q7 –Q8

The results were grouped for the 'controlled arm' (sense of ownership or agency towards the side a participant controlled) and 'non-controlled arm' (sense of ownership or agency towards the side that was not controlled by the individual). Figs 3 and 4 summarize the results of the questionnaires.

We calculated the sense of body ownership towards the controlled and non-controlled arms of the virtual joint avatar in the Tied and Separated conditions (Fig 3). A two-way repeated measures ANOVA with ART (aligned rank transformation) procedure [49] was conducted since the data significantly deviated from normality according to the results of Shapiro-Wilk tests (controlled arm for Tied: W = .718, p < .001, non-controlled arm for Tied: W = .927, p = .085, Controlled arm for Separated: W = .851, p = .002, non-controlled arm for Separated: W = .892, p = .015). There was a significant interaction between the condition and the arm ($F_{(1,23)} = 28.457$, p < .001, $\eta_p^2 = 0.55$). Simple main effects analysis showed that the ownership for the non-controlled arm in the Tied condition was significantly higher than the ownership for the non-controlled arm in the Separated condition ($F_{(1,23)} = 11.837$, p = .002, $\eta_p^2 = 0.34$). In both Tied and Separated conditions, the ownership for the controlled arm was higher than the ownership for the non-controlled arm (Tied: $F_{(1,23)} = 108.62$, p < .001, $\eta_p^2 = 0.825$, Separated: $F_{(1,23)} = 111.3$, p < .001, $\eta_p^2 = 0.829$).

### Sense of agency towards the non-controlled arm became less negative in Tied condition over Separated condition

We calculated the sense of agency towards the controlled and non-controlled arms of the virtual avatar in the Tied and Separated conditions (Fig 4). A two-way repeated measures ANOVA with ART (aligned rank transformation) procedure [49] was conducted since the data significantly deviated from normality according to the results of Shapiro-Wilk tests (controlled arm for Tied: W = .829, p < .001, non-controlled arm for Tied: W = .942, p = .183, Controlled arm for Separated: W = .890, p = .013, non-controlled arm for Separated: W = .793, p < .001). There was a significant interaction between the condition and the side ($F_{(1,23)} = 13.095$, p = .0014, $\eta_p^2 = 0.363$). Simple main effects analysis showed that the agency for the non-

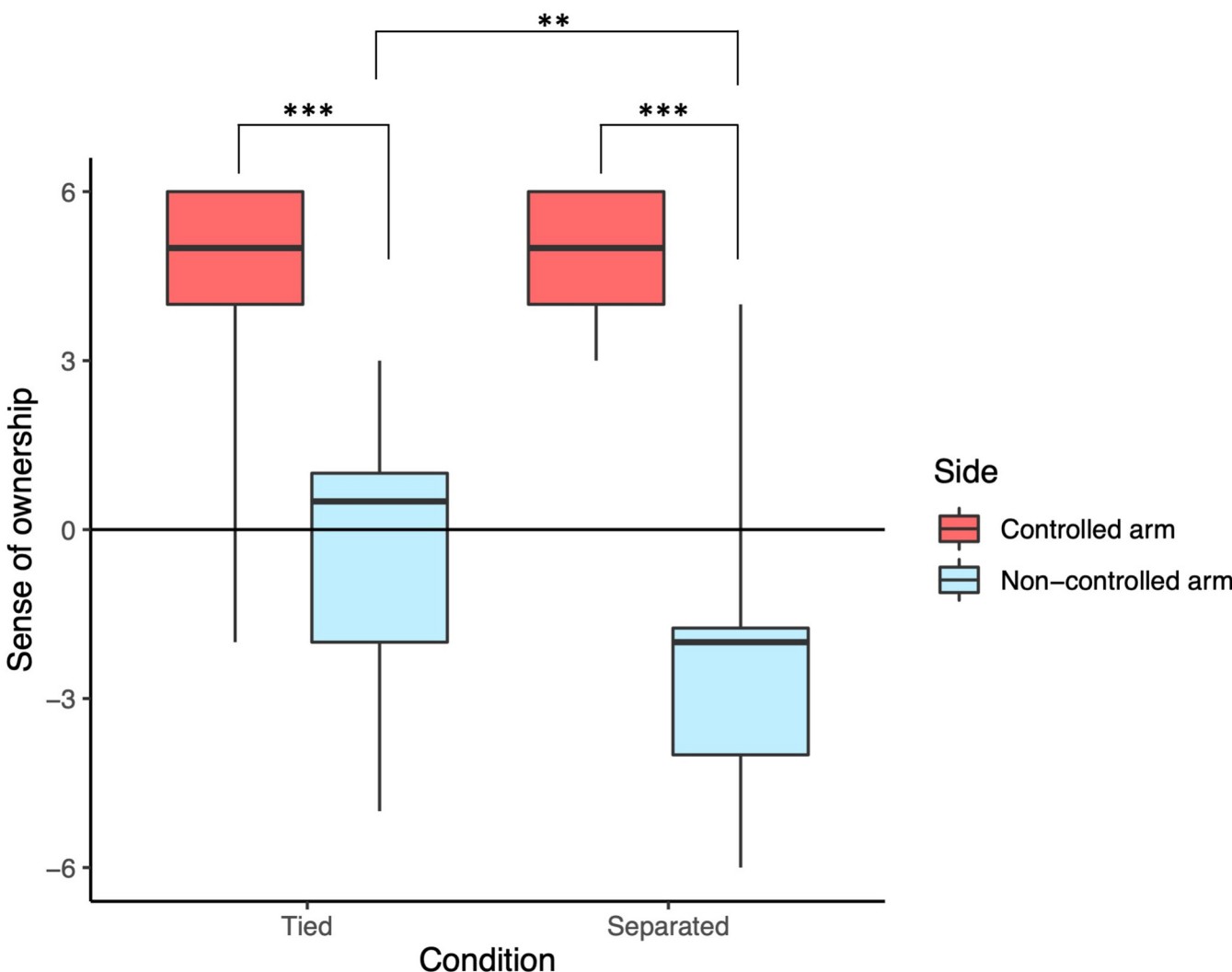

**Fig 3. Sense of body ownership towards the controlled and non-controlled arms of the joint virtual avatar in Tied and Separated conditions.** The ownership for the non-controlled arm in the Tied condition was significantly higher than the ownership for the non-controlled arm in the Separated condition. In both Tied and Separated conditions, the ownership for the controlled arm was higher than the ownership for the non-controlled arm.

controlled arm in the Tied condition was significantly higher than the agency for the non-controlled arm in the Separated condition (F(1,23) = 16.512, p < .001, $\eta_p^2$ = 0.418). In both Tied and Separated conditions, the agency for the controlled arm was higher than the agency for the non-controlled arm (Tied: F(1,23) = 106.94, p < .001, $\eta_p^2$ = 0.823, Separated: F(1,23) = 121.76, p < .001, $\eta_p^2$ = 0.841).

### Proprioceptive drift towards the non-controlled shoulder occurred only after the Tied condition

In this experiment, we conducted three behavioral tests to quantify possible changes in the body representation that may accompany the sense of ownership towards the joint avatar. These tests were concentrated on testing the participant's spatial perception of their shoulder above the non-controlled arm (non-controlled shoulder), which was purposely shown displaced by 10 cm (away from the head, and along the axis of the participant's shoulder) from

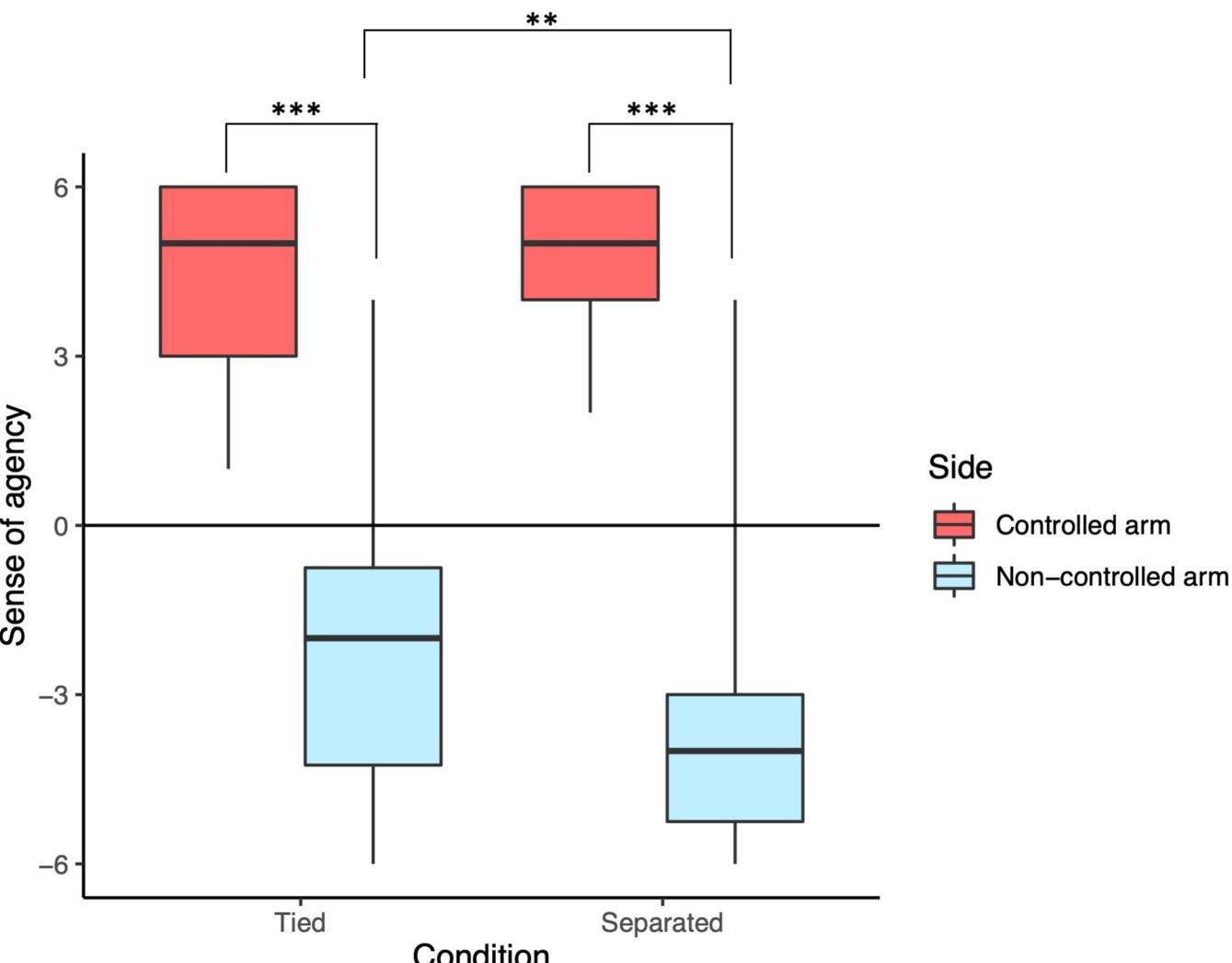

**Fig 4. Sense of agency towards the controlled and non-controlled arms of the joint virtual avatar in Tied and Separated conditions.** The agency for the non-controlled arm in the Tied condition was significantly higher than the agency for the non-controlled arm in the Separated condition. In both Tied and Separated conditions, the agency for the controlled arm was higher than the agency for the non-controlled arm.

the actual location of the participant's shoulder. We hypothesized the spatial perception of the non-controlled shoulder by the participants to be affected differently in the Tied and Separated conditions. The three tests (see inset images in Fig 5 for intuition) were carried out to test the perceived position of the real shoulder of the non-controlled side using a pointer in test 1, perceived real shoulder width in test 2, and perceived position of the real non-controlled side shoulder pointed with the real controlled arm in test 3.

Fig 5 shows the results of the three tests in the pre and post tasks that were carried out to measure the drifts in proprioception after experiencing the reaching task with a non-controlled shoulder drift of 10 cm towards the outer side of the body. In Fig 5, positive values indicate a higher drift towards the virtual avatar's shoulder position in the post-test than in the pre-test.

Since Shapiro-Wilk tests showed that the Separated condition deviated from normality (Tied: W = .976, p = .808, Separated: W = .885, p = .011), Wilcoxon signed rank tests were performed for the data of Test 1. Wilcoxon signed rank test showed that the difference in drifts in

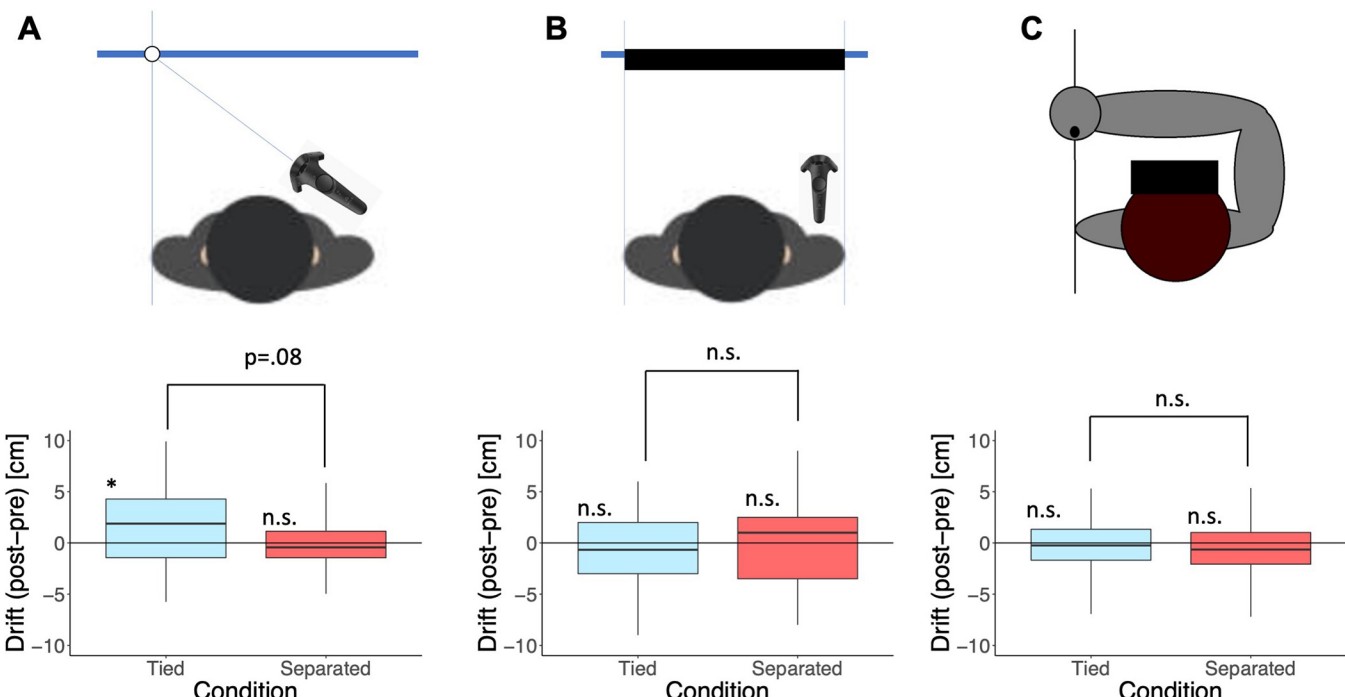

**Fig 5. Behavioral tests. (A)** The pointed virtual shoulder position drifted significantly larger than 0, while the mean in the Separated condition was not significantly different from 0, suggesting that there was a drift towards the virtual shoulder position only after the Tied condition. However, the difference in drift between the two conditions did not reach significance. No significant changes were observed in **(B)** Test 2, and **(C)** Test 3.

the post and pre-tests towards the virtual shoulder position was nearly significantly higher in the Tied condition compared to that in the Separated condition (W(23) = 200.000, p = .080, effect size r = 0.333). One sample tests (Student t-test for Tied and Wilcoxon signed rank test for Separated) showed that the mean in the Tied condition was significantly larger than 0 while the mean in the Separated condition was not significantly different from 0 (Tied: t(23) = 2.134, p = .022, d = 0.436, Separated: W(23) = 130.000, p = .584, effect size r = -0.133), meaning that there was a drift towards the virtual shoulder position only after the Tied condition.

Results of Test 2 showed a normal distribution (Tied: W = .966, p = .342, Separated: W = .970, p = .671; Shapiro Wilk test). A paired samples t-test showed that the difference in drifts in the post and pre-tests towards the virtual shoulder position was not significantly different in the Tied condition compared to that in the Separated condition (t(23) = -.303, p = .765, d = -0.062; Fig 5B). One sample t-tests showed that the means of both conditions were not significantly different from 0 (Tied: t(23) = -1.016, p = .32, d = -.207, Separated: t(23) = -.417, p = .681, d = -.085).

Results of Test 3 also showed a normal distribution (Tied: W = .982, p = .927, Separated: W = .988, p = .991; Shapiro Wilk test). A paired samples t-test showed that the difference in drifts in the post and pre-tests towards the virtual shoulder position was not significantly different in the Tied condition compared to that in the Separated condition (t(23) =, p = .921, d = 0.020; Fig 5C). One sample t-tests showed that the means of both conditions were not significantly different from 0 (Tied: t(23) = -.746, p = .463, d = -0.152, Separated: t(23) = -.843, p = .408, d = -0.172).

## Discussion

In this study, we investigated whether just the upper-body movements induced by the movements of a virtual arm controlled by another, can affect/enhance embodiment towards it. We

developed a joint avatar for this investigation, in which the left and right-limbs were controlled by two different individuals. We compared the senses of body ownership and agency towards the two arms (the controlled arm and the non-controlled arm) of the joint virtual avatar under *Tied* (with connection induced synchronized upper body movements) and *Separated* (without passive synchronized upper body movements) conditions. The results showed that the sense of body ownership towards the non-controlled arm was significantly less negative when the participants were provided connection induced upper body movements synchronized with the movements of the partner-controlled arm during the reaching task compared to when they were not provided any passive movements. Furthermore, the sense of agency towards the non-controlled arm was also less negative when passive synchronized upper body movements were present. Thus, H1 was supported, but the effects of passive synchronized upper body movements or connection force were not large and limited.

To the best of the authors' knowledge, this is also the first study to investigate the concept of 'joint avatar' and corresponding 'joint ownership', in which two individuals own an avatar with separate limbs controlled fully by each. To quantify the sense of ownership and agency during co-embodiment, in addition to questionnaires, we utilized three behavioral tests where we tried to measure possible changes in the body representation in the participants when they embodied the joint avatar. Tests 1 and 2 are related to body image since these tests quantified the participant's perception of their body (specifically location and width of the shoulder in our case) in the visual space. On the other hand, our Test 3 tried to analyze changes in *body schema*, which refers to the internal (non-visual) representation of one's body in the proprioceptive and haptic space. In Test 3, participants had their HMDs blacked out (similar to eyes closed) and they used their sensory-motor function of the controlled side arm to point at the shoulder position making test 3 related to body schema. We conducted three types of tests to ensure no changes in body representation goes unobserved due to possible limitations of any one of the tests. While we could observe tendencies of changes in the localization of the non-controlled shoulder in Test 1 (in which the localization was done with a pointer), the difference between the tied and separated conditions was not significant. Thus, H2 was partly supported, but we need further discussion and future investigation.

Proprioceptive drifts are known to occur in both active (participant moves the rubber hand) and passive (experimenter simultaneously moves the rubber hand and the participant's corresponding hand) rubber hand illusions (in the spatially congruent condition of Kalckert & Ehrsson (2012) [42]), suggesting that similar levels of ownership are induced regardless of whether the movements were active or passive when the rubber hand is presented in an anatomically congruent posture [42]. The proprioceptive drift of the non-controlled limb in the Tied condition of Test 1 in our experiment is somewhat consistent with this and interestingly suggests that a certain level of ownership can be induced to limbs even by passive synchronous movements of the upper body (instead of the limb itself).

However, we did not observe any body-representation changes in Test 2 (localization of the shoulder width) and Test 3 (localization of non-controlled shoulder with the controlled hand). Thus, the results of the behavioral tests were not conclusive. This may be due to a several reasons. First, the passive upper body movements were not natural enough. While our back brace setup could provide appreciable passive movement, it had its limitations. The brace provides a forward push on the non-controlled shoulder and a backward pull on the controlled shoulder creating an overall rotational movement of the upper body. This is different from the pure forward pull that would be expected on the non-controlled shoulder if the non-controlled arm was really fixed to one's body. Furthermore, while we tried to strategically place the targets so that the participants had to twist their bodies (and hence provide a force and upper body postural feedback to their partner) to reach them, we observed that some participants could still

reach some targets without much movement of the upper body. We speculate these limitations in the setup limited the induced sense of ownership in our experiment. To test this speculation, using the body position data of participants recorded during the reaching task, we calculated the distances moved by the real shoulder of participants in the non-controlled side each time the partner performed the reaching in the Tied condition. The average movement of the shoulder (how much the shoulder was pushed by the partner in average) for all participants was calculated and we checked the correlation of this variable with ownership of the non-controlled arm, agency of the non-controlled arm, and drifts in tests 1, 2, and 3 in the tied condition. However, no correlations were seen (Ownership: $r(22) = .095$, $p = .330$, Agency: $r(22) = .057$, $p = .396$, Test 1: $r(22) = -.126$, $p = .724$, Test 2: $r(22) = .275$, $p = .097$, Test 3: $r(22) = -.268$, $p = .898$). Therefore, this exploratory correlation analysis between the amount of movement and the subjective/behavioral results does not support our speculation.

The second reason for insignificant results in tests 2 and 3 maybe the fact that we used the same order of tests; All participants did Test 1 first, followed by Test 2 and Test 3, with each test taking a few minutes, and (related to the first reason) possible changes in body representation probably decreased over time. And finally, in Test 3, our measure of participant shoulder localization may have been affected by the possible uncertainties participants had regarding the marker (that they point to with their controlled arm) position on their shoulder. Overall, these observations suggest that the use of better tests may be required to measure change of body representation due to embodiment of joint avatars.

Previous studies have shown the requirement of multi-sensory stimulation and/or agency to be a key 'bottom up' requirement for the sense of ownership [31, 50]. Popularly these studies have utilized visuo-haptic or visuo-proprioceptive stimulation that are spatially coincident in the vision space (e.g., [29, 37, 41]). That is, the haptic or proprioceptive sensation comes from the arm you are observing. We believe that this is very different from the feedback in the present study. This is because the passive movement is felt not on the embodied limb but on the shoulders or upper body. Even though the passive movement is already known to induce ownership of artificial limbs [38, 41, 42], in those previous studies the experimenter has passively moved an artificial finger connected to the real finger of the participant and shown that it induces ownership towards the artificial finger. The sense of agency towards the passively moved finger was less negative when the actual hand and the rubber hand were placed anatomically congruent than when they were incongruent [42]. If we match these to our experiment, it will be similar to a scenario where the real arm of the participant in the non-controlled side is passively moved (by the partner or the experimenter) along with the virtual limb. However, instead here we provide the participants with only the connection induced movements and forces on their shoulder and upper body. Therefore, our study focuses on passive movement of the upper body (or the place where the virtual limb is supposed to join the body) as a result of limb movements by the partner. This result has important ramifications for the understanding limb embodiment, as it suggests physical connection and/or postural change of upper body as a factor affecting the senses of ownership and agency. Our results can provide new insights into the perception of prosthetic limbs or robotics supernumerary limbs that are physically attached to a user's body, even in the absence of controllability and sensory feedback, or even when the device is passive.

Previous research has shown that verbal information previewing movements of another's arms placed to seem like a person's own arms can induce vicarious agency to those arms [45]. While it is possible that the passive upper body movements conveyed cues previewing the movements of the partner's arm to some degree, the verbal information previewing movement is more cognitive and detailed (for example, "wave hello with your right hand") compared to the passive upper body movements in our study that informs only the movement timing,

direction, and force. Furthermore, the targets reached by the partner were invisible to participants in our experiment making it harder to guess exactly where in space the partner-controlled arm would move. Therefore, influence from previewing information regarding the arm movements were controlled as possible in our experiment. If previewing verbal information about arm movements are also provided along with the passive upper body movements we can expect an even higher score for sense of agency for partner-controlled limbs. This has to be explored in future studies.

Since the differences between Tied and Separated conditions are clearly recognizable to the participants, one may argue that the differences of the sense of ownership and agency between these conditions could be caused by higher-order cognitive comparisons and not by the difference of perceptual embodiment itself. If so, the effect of the order would be large. To test this possibility, we performed an exploratory analysis separating the data into two different ordered groups and compared them. If the comparison between the conditions as a between participant design showed a difference irrespective of the first or second session, it supports that the significant difference between conditions found in this study were caused by the strength of the sense of ownership and agency themselves and not by the relative difference by some higher-order cognitive judgment. On the other hand, if there was no effect in the first half session and a clear effect in the latter half, the significant difference may possibly be caused by cognitive comparisons. We performed a mixed-design three-way ANOVA-ART [49] with the condition order (ST: Separated condition first, Tied condition second, TS: Tied condition first, Separated condition second) as a between-subject variable, the condition (Separated condition, Tied condition) as a within-subject variable, and the side (Controlled arm, Non-controlled arm) as a within-subject variable for both the senses of ownership and agency. For the sense of ownership, we found main effects of the condition ($F(1,22) = 10.015$, $p = .004$, $\eta_p^2 = 0.315$) and the side ($F(1,22) = 131.71$, $p < .0001$, $\eta_p^2 = 0.857$), and an interaction between the condition and the side ($F(1,22) = 26.605$, $p < .0001$, $\eta_p^2 = 0.507$), consistent with the original 2-way ANOVA. We did not find a 3-way interaction between condition, side and order ($F(1,22) = 0.277$, $p = .604$, $\eta_p^2 = 0.012$), suggesting that the condition order did not affect the ownership towards the non-controlled arm in the Tied condition over the Separated condition. We found an interaction between the order and condition ($F(1,22) = 6.91$, $p = .015$, $\eta_p^2 = 0.239$). However, no simple effects were significant, and the difference between Tied and Separated conditions was larger when the Separated condition was performed first than when the Tied condition was performed first (ST: $F(1,35) = 2.339$, $p = .135$, $\eta_p^2 = 0.063$, TS: $F(1,35) = 0.115$, $p = .736$, $\eta_p^2 = 0.003$). For the sense of agency, we found main effects of the condition ($F(1,22) = 10.537$, $p = .004$, $\eta_p^2 = 0.324$) and the side ($F(1,22) = 134.37$, $p < .0001$, $\eta_p^2 = 0.859$), and an interaction between the condition and the side ($F(1,22) = 16.343$, $p = .0005$, $\eta_p^2 = 0.426$), consistent with the original 2-way ANOVA. We did not find a 3-way interaction between condition, side and order ($F(1,22) = 0.331$, $p = .571$, $\eta_p^2 = 0.015$), suggesting that the condition order would not affect the agency towards the non-controlled arm in the Tied condition over the Separated condition. We did not find any other effects.

To test the effect of order further in detail, we separated the data into the first session and the second session (accordingly the force condition (Tied/separated) is between-subject in this analysis) and performed another exploratory analysis. We performed a mixed-design two-way ANOVA-ART with the condition (Separated condition, Tied condition) as a between-subject factor, and the side (Controlled arm, Non-controlled arm) as a within-subject factor for both senses of ownership and agency. Analyses were performed separately for the first session and the second session. For the sense of ownership, we found a main effect of the side ($F(1,22) =$

99.33, p < .0001, $\eta_p^2$ = 0.819) and an interaction between condition and side (F(1,22) = 4.833, p = .0387, $\eta_p^2$ = 0.180) for the first session, similar to the original analysis. However, the simple effect analysis showed that the condition effect on the sense of ownership towards the non-controlled arm in the Tied condition compared to the Separated condition was not significant (F(1,22) = 2.490, p = .129, $\eta_p^2$ = 0.102). This may be due to the data division. In the second session, we obtained main effects of condition (F(1,22) = 5.350, p = .030, $\eta_p^2$ = 0.196) and side (F(1,22) = 89.037, p < .0001, $\eta_p^2$ = 0.802), but the interaction between condition and side was not significant (F(1,22) = 3.912, p = .0606, $\eta_p^2$ = 0.151) for the second session. For the sense of agency, we found a main effect of the side (F(1,22) = 116.49, p < .0001, $\eta_p^2$ = 0.841), but the interaction between condition and side was not significant (F(1,22) = 2.680, p = .1159, $\eta_p^2$ = 0.109) for the first session. In the second session, we obtained main effects of condition (F(1,22) = 5.433, p = .0293, $\eta_p^2$ = 0.198) and side (F(1,22) = 99.766, p < .0001, $\eta_p^2$ = 0.819), but the interaction between condition and side was not significant (F(1,22) = 3.278, p = .0839, $\eta_p^2$ = 0.130) for the second session. To summarize, we cannot conclude that the effect of order never affects the results. However, the results cannot be solely explained by the higher-order cognitive bias because the effect of order was small, and the interaction between the condition and side was significant in the first session, where participants did not know both Separated and Tied conditions.

## Limitations of the study

While we could observe clear cognitive differences in the feeling of ownership and agency due to the introduction of connection induced upper body movements (in the tied condition), the three tests we chose to measure behavioral differences were inconclusive. Apriori, we did not conduct physiological tests like skin conductance variations in response to a threatening stimulus [51], hand temperature [52] and somatosensory evoked potentials [53] due to the presence of movements in our task, which significantly reduce the signal to noise ratio of these tests. But these and other new behavioral tests are needed in future studies to assess the effects of connection induced movements on embodiment.

To simulate passive movements on the upper body felt due to a real prosthetic arm being moved, our brace device with support poles provided pushing and pulling forces to the non-controlled shoulder. However, pulling and pushing forces had to be applied to the controlled shoulder as well for this setup to work and provide an overall rotational movement to the upper body since the reaching task involved rotational movements of the upper body to reach targets far away from the avatar. Although these forces to the controlled side makes the simulation deviate from real upper body movements that may be caused by a self-moving/autonomous prosthetic arm, these pulling/pushing forces were necessary for our setup to successfully provide upper body movements to each other in synchrony with the virtual arm controlled by the partner. A more sophisticated force feedback device like a robot exoskeleton setup, may have to be used to simulate a more realistic haptic feedbacks closer to upper body movements caused by the pull of a moving attached autonomous arm and minimize such limitations.

Furthermore, with the current setup, we cannot dissociate the contributions of postural and haptic feedbacks to the results since postural changes were correlated and in fact caused by the connection forces. Thus, it is an open question whether the haptic feedback alone or the postural feedback of the shoulder alone can affect the sense of embodiment. This should be investigated in a future study.

## Methods

### Participants

24 male participants (mean/S.D. age: 23.6±1.8) took part in the experiment. Participants were recruited from the university and were naïve with respect to the purpose of the experiment and had normal or corrected-to-normal vision. The individuals pictured in S1 Movie have provided written informed consent (as outlined in PLOS consent form) to publish their images alongside the manuscript. All volunteers provided written informed consent before the experiment. This study was approved by the Ethical Committee on Human-Subject Studies of Toyohashi University of Technology, and all methods were carried out in accordance with the relevant guidelines and regulations. The sample size was determined by a power analysis with an effect size (f) of 0.25 (medium), an alpha of 0.05, and power of 0.80 using G*Power 3.1 [54, 55].

### Setup and apparatus

The participants took part in the experiment as dyads. Each dyad was chosen in a way that the height difference of the two participants was never more than 5 cm. The movements of the two participants were measured by a motion-capture system (Vicon Bonita10, 12 cameras, 1024 x 1027 pixels, 250 fps, focal length: 4–12 mm, F/1.4-CLOSE, angle of view: 26.41 x 26.41 deg), and processed in a computer (HP Z440 Workstation, OS: Windows7, CPU: Intel(R) Xeon E5-1620 v3, 3.5GHz, RAM: 32GB, GPU: NVIDIA Quadro 5000 (2560MB GDDR5)) with middle-level software (Vicon Blade 3.4.1, Vicon Pegasus 1.2.2). Two computers with same specifications (DELL XPS 8930, OS: Windows10, CPU: Intel(R) Core i7-9700 3.0GHz RAM: 32GB, GPU: NVIDIA GeForce RTX 2060 SUPER) received the processed information of the motion capture data and presented the virtual environment on head mounted displays (HMD: HTC VIVE Pro Eye, 1440 x 1600 pixels per eye, 90 x 110deg, 90 Hz refresh) of both participants. The virtual environment was created using Unity (2017.4.1f1).

### Stimuli and conditions

The experiment was conducted in a virtual environment using a male avatar of height 175 cm (hand length 62 cm). The full body movements of the two participants (excluding the finger movements) were read using a Vicon motion capture system at a rate of 250 fps and were reflected on to the joint avatar in a way that one participant controlled the left side limbs (left arm and the left leg) of the joint avatar (left participant), while the other participant controlled the right side of the avatar body (the right participant). The position of the joint avatar (center bone/spine) was calculated to be the mean position of the two participants. Both participants were allowed to freely move their heads and look around in the virtual environment from the first-person perspective of the joint avatar through their Head-Mounted displays (HMDs).

The experiment was conducted under two conditions as shown in Fig 2. Here the movements of the partner were read using the motion capture system and rotated by 180˚ in unity to match the direction of the joint body since the participants faced opposite directions during the experiment.

In the tied condition, the two participants were asked to stand leaning on to two 'support poles' set behind them (see Fig 2A and 2C). They were connected to each other's upper body backs using two solid rectangular frames (back braces) as shown in Fig 2A. These back braces were pasted to the bodies of the participants with hook and loop fastener and also tied around the shoulders with elastic bands to fit their bodies and make firm connections.

This set up induced passively moved the participants to adjust their upper body postures to match the posture of the joint avatar experienced in VR. Each support pole acted as an axis

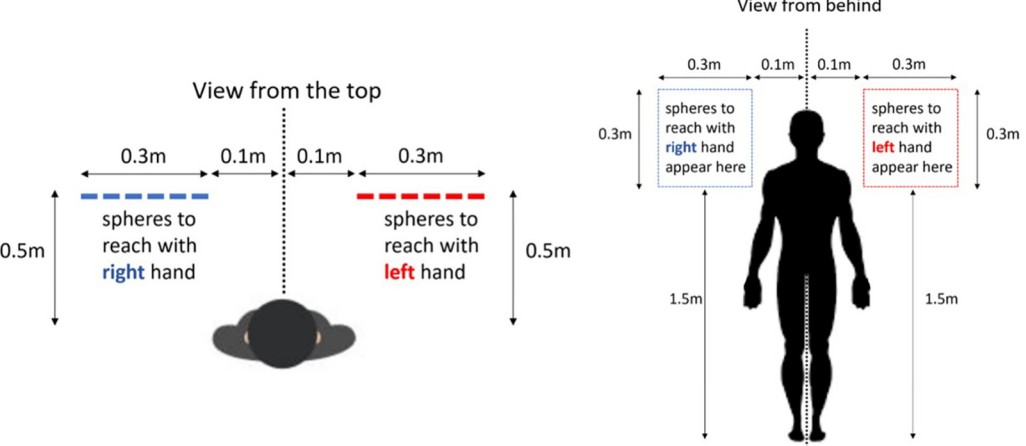

**Fig 6. Area in which targets appeared in the VE for participants controlling left (red) and right (blue) sides.** Targets appeared inside the 2D squares shown by dotted traces. Targets for the right participant appeared towards the left side of the joint avatar and the targets for the left participant appeared towards the right side of the joint avatar. All targets that appeared inside the blue square were symmetrical to the ones that appeared in the red square to balance the amount of work done by each participant.

around which the participants could rotate their upper bodies during the exchange of forces between each other. The targets were set to appear at positions that required the participants to change their upper body posture and reach further than their hands can reach alone towards the targets in order to reach them (See Fig 6 for the target positions).

In the separated condition, each participant was connected to a mannequin set to their partner's height in the same way they were connected to their partner in the tied condition (see Fig 2B and 2D). The posture of the mannequin could be changed through forces conveyed to it through the back braces by the participants. Both participants were connected to mannequins while in VR they performed the same reaching task as in the tied condition. Therefore, in the separated condition, the participants' postures were not affected by their partners' movements

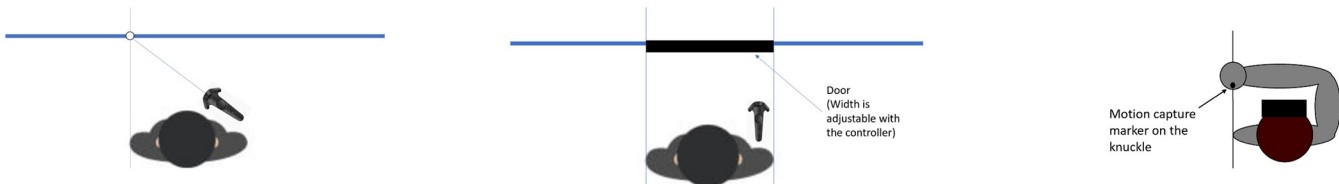

**Fig 7. Test 1—Visually measuring the position of the real shoulder. A.** A wall of height 3m appeared 1m in front of each participant parallel to their bodies (shoulders) and the avatar was made invisible. On the wall, a horizontal straight line appeared at a random height within the range of ±10 cm vertically from the shoulder position of the participant (shoulder height was read from the motion capture marker pasted on the shoulder of the participant). A controller was handed to the controlled hand of each participant, and they were asked to point on the line, the position that corresponds to the edge of their non-controlled shoulder. Here the figure shows the ideal case in which the right participant points perfectly at the edge of their left shoulder. The task was opposite for the left participant, who had to use his left hand to point to his right shoulder. **B. Test 2—Measuring the width of the real body.** A wall of height 3m appeared 2m in front of the participants and the avatar was made invisible. On the wall a door of height 2m and width 30 cm appeared within the range of ±10 cm horizontally from the center of the body. Using the touch pad buttons of the controllers given to each participant, they were asked to adjust the width of the door to match the width of their real bodies. This figure demonstrates a perfectly matched scenario. **C. Test 3—Measuring the proprioceptive position of the shoulder.** During this test, the HMD was blacked out to avoid the participants seeing anything in VR or their real bodies. They were asked to move their controlled arm hand to place the motion capture marker on the knuckle of that hand just in front of the end of the shoulder of the non-controlled arm shoulder of their real bodies (participants were asked to remember the position of the specific motion capture marker pasted on the knuckle of the controlled arm before the experiment started). They were instructed to make the localization in one smooth movement and not to touch/search for the shoulder. This figure demonstrates a perfectly pointed scenario.

during the reaching task. However, they could give force feedback to the mannequin while reaching as they did in the tied condition.

The left shoulder position of the virtual joint avatar viewed by the right participant was shifted 10 cm away from the head during both conditions such that the avatar shoulder was wider than the real shoulders of the participant. Similarly, the right shoulder position of the joint avatar viewed by the left participant was shifted 10 cm away from the avatar head. This shift was used to measure changes in the participant's perception of their real shoulder position.

The real arm of the participant that required no movement (non-controlled arm) was fixed on the motion capture suit with magic tape and the participants were asked to relax and not use that arm during the experiment. This was to ensure that the participants do not move the real arm and therefore possibly minimize the mismatches of sense of location of the perceived virtual arm controlled by the partner and the real arm of the participant.

In each condition, the participants performed 100 trials (reaches) as a team (50 reaches each). In each trial one of the participants was presented with a target in front of him and was required to reach the target as quickly as possible. The target was a sphere of 5 cm that appeared at a random position in the virtual environment at a reachable distance with the hands of the joint avatar. The target disappeared 200 ms after the tip of the index finger of the joint avatar reached it (at the instant where the velocity of the finger first dropped below 1 cm/s roughly within the volume of the sphere). 2 seconds later the next target appeared. The reach movement made by the joint avatar was however visible to both participants in first person perspective. Both participants had total control over the head movements (in first person perspective) separately even though one half/side of the avatar body below the neck was totally controlled by the partner.

The target for the participant controlling the right side of the joint body, always appeared in front of the left side of body, and vice-versa for the participant controlling the left side of the joint body (Fig 6). This ensured that a participant would have to rotate his shoulders around the support pole pulling the left shoulder of the partner and pushing the right shoulder of the partner through the back braces. This enables the partner to receive a connection induced upper body movement synchronized with the movement of the joint avatar he views (where he will see the right arm moving forward while experiencing a push from behind to his right shoulder synchronized with the movement of the virtual limb). Similarly, when the target appeared to the participant controlling the left half of the joint avatar, it always appeared in front of him towards his right making it necessary for him to provide a synchronized passive movement to the shoulders of his partner when he reaches for the target. The participants were asked to relax their bodies and let their partners pull/push their shoulders during the task when it was not their turn to perform the reaching and were asked to observe the movements of the virtual limb controlled by their partner. We instructed them to move the shoulders and reach for the targets as necessary when it was their turn to reach, since it makes the reaching easier and also provides a well synchronized passive movement to the upper body of the partner.

## Procedure

Behavioral tests to measure embodiments were performed before and after the reaching task as pre and post-test tasks. We carried out three tests, two (Test 1 and Test 2) that quantified changes in the body image related to the participant's shoulder, and one (Test 3) that estimated the body schema related to the position of the participant's shoulder.

We shifted the position of the shoulder controlled by the partner 10 cm away from the head of the virtual avatar (participants were not informed of this shift). We hypothesized that the

answered positions of a participant in the three tests would drift more towards the outer side of the body (after the reaching task compared to before) if the avatar is embodied by the participant.

The three tests were carried out in the order of test 1, test 2, and test 3 (Fig 7). In Test 1, the position of the real shoulder of the non-controlled arm was measured by visual pointing on the front-parallel plane in front of the participants (Fig 7A). A wall of height 3m appeared 1m in front of each participant parallel to their bodies (shoulders) and the avatar was made invisible. On the wall, a horizontal straight line appeared at a random height within the range of ±10 cm vertically from the shoulder position of the participant (shoulder height was read from the motion capture marker pasted on the shoulder of each participant). A controller was handed to the controlled hand of each participant, and they were asked to point on the line, the position that corresponds to the edge of their non-controlled shoulder. The participants confirmed their location answer by pulling the trigger button on the controller. When the answer was detected, the screen was made to black out for 2 seconds and the next test was started.

In Test 2, the width of the real shoulder was measured by adjusting an aperture width in front of the participants (Fig 7B). A wall of height 3m appeared 2m in front of the participants and the avatar was made invisible. On the wall a door of height 2m and width 30 cm appeared within the range of ±10 cm horizontally from the center of the body. Using the touch pad buttons of the controllers given to each participant, they were asked to adjust the width of the door to match the width of their real bodies. The participants confirmed their answers by pulling the trigger button of their controllers and the width of the door at that point was recorded. When the answer was detected, the screen was made to back out for 2 seconds and the next test started.

In Test 3, the perceived proprioceptive position of the real shoulder of the non-controlled arm was measured by asking the participants to point at their non-controlled shoulder with their controlled hand (Fig 7C). During this test, the HMD was blacked out to avoid the participants seeing anything in VR or their real bodies. They were asked to move their controlled hand to place the motion capture marker on the knuckle of that hand just in front of the end of the shoulder of the non-controlled shoulder of their real bodies (participants were asked to remember the position of the specific motion capture marker pasted on the knuckle of the controlled arm before the experiment started). They were instructed to make the localization in one smooth movement and not to touch/search for the shoulder. The participants confirmed their answers by pulling the trigger button on the controller and the horizontal distance between the real shoulder end of the non-controlled arm hand and the answered position was recorded.

Each test was carried out 3 times in the order of test 1, test 2, test3 and the median value for each test from the 3 repetitions was used for analysis of drift in each test.

There were two sessions in the experiment, one trial of 5–7 minutes (depending on the response times of the participants) in the *Tied* condition and one trial of similar length in the *Separated* condition (Fig 8). Each session consisted of the pre-test task, the reaching task, and the post-test task explained above followed by a questionnaire in the end. Half the dyads started with the *Tied* condition while the other half started with the *Separated* condition (to eliminate any influence by experiencing either of the conditions first).

During the reaching task, the participants were asked to reach a target sphere as quickly as possible. The target disappeared 200ms after the reaching, and another target appeared at a different position. The target appeared 5 times consecutively for each participant (to provide enough time for the partner to feel his virtual hand being controlled and reduce the chances of being too focused on the target and losing focus on the virtual hand controlled by the partner and the passive upper body movements from the non-controlled arm). After 5 reaches, a

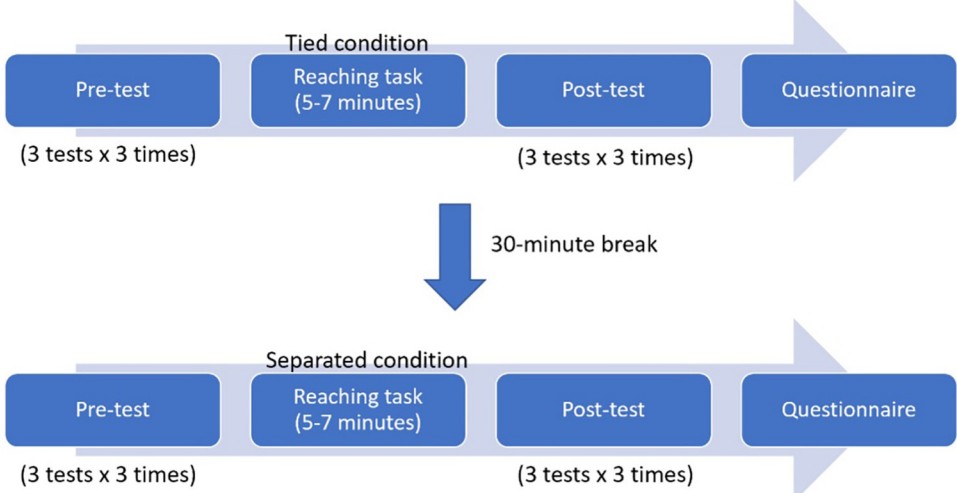

**Fig 8. Design flow of the experiment.** First, the participants performed the three types of proprioception tests mentioned in Fig 7 to answer the location of their real non-controlled shoulder/width of their body before the reaching task. After this pre-test, the reaching task was performed for 5–7 minutes (100 target reaches in total) and another session similar to the pre-test was carried out soon after the reaching task finished to calculate the proprioception towards the real non-controlled shoulder. After that, the participants answered a questionnaire of 8 questions which we used to calculate senses of agency and ownership. After a 30-minute break, they performed another similar session in the next condition (Tied or Separated).

recorded voice instruction was played through the HMD headphones of both participants saying "change", meaning it is the turn for the other participant to perform the reaching 5 times (using the other limb of the joint avatar). This way, 20 sets of 5 reaches were performed in the order L, R, L, R (where L: appeared to the left participant, and R: appeared to the right participant). Therefore, each participant performed 50 reaches and observed 50 movements by their partner during one session.

The participants were instructed to relax their bodies and let the passive movements from the non-controlled arm guide their body movements while observing the movement of the virtual hand in the Tied condition. In both the Tied and Separated conditions, both participants were awarded points during the reaching task as a team (0 points if the response time was longer than 1.5 seconds, in other cases, if the target error was less than 3 cm: 30 points, if the target error more than 3 cm but less than 10 cm: 15 points). The total score was displayed on the wall in front of the avatar and each time the target was reached, the number of points gained was announced through the headphones of the HMDs of both participants using pre-recorded voice recordings (e.g.- "zero", "thirty" etc.). This was done to motivate the participants to work as a team and perform the reaching as quickly as possible with little reaching errors.

In between the two sessions, we gave the participants a break of 30 minutes. At the end of the session, the participants were asked to answer a questionnaire regarding how they felt about the left and right hands of the joint virtual avatar during the reaching task. Questionnaire consisted of 8 items, 4 regarding the left hand and 4 regarding the right hand. Participants answered all the questions.

Q1. I felt as if the virtual left hand I saw was my left hand

Q2. It felt as if the virtual left hand I saw was someone else

Q3. It felt like I could control the virtual left hand as if it were my own left hand

Q4. I felt as if the virtual left hand was moving by itself

Q5. I felt as if the virtual right hand I saw was my right hand

Q6. It felt as if the virtual right hand I saw was someone else

Q7. It felt like I could control the virtual right hand as if it were my own right hand

Q8. I felt as if the virtual right hand was moving by itself

Questionnaire answers were performed in a 7-point Likert scale from -3 to +3 where -3 meant "Strongly disagree" and +3 meant "Strongly agree".

## Supporting information

**S1 Movie. Outline of experimental methods.** The left and right halves of the avatar were controlled by different participants, who were connected to or separated from each other. (MP4)

## Author Contributions

**Conceptualization:** Harin Hapuarachchi, Gowrishankar Ganesh, Michiteru Kitazaki.

**Funding acquisition:** Gowrishankar Ganesh, Michiteru Kitazaki.

**Investigation:** Harin Hapuarachchi.

**Methodology:** Harin Hapuarachchi, Takayoshi Hagiwara, Gowrishankar Ganesh, Michiteru Kitazaki.

**Supervision:** Gowrishankar Ganesh, Michiteru Kitazaki.

**Writing – original draft:** Harin Hapuarachchi, Gowrishankar Ganesh, Michiteru Kitazaki.

**Writing – review & editing:** Harin Hapuarachchi, Gowrishankar Ganesh, Michiteru Kitazaki.

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
