## [Decision Letter · Decision Letter 0]

21 Jun 2022

PONE-D-21-38910Connection force feedback in joint avatar increases embodiment towards a body part controlled by anotherPLOS ONE

Dear Dr. Hapuarachchi,

Thank you for submitting your manuscript to PLOS ONE. After careful consideration, we feel that it has merit but does not fully meet PLOS ONE’s publication criteria as it currently stands. Therefore, we invite you to submit a revised version of the manuscript that addresses the points raised during the review process.

We look forward to receiving your revised manuscript.

Kind regards,

Josh Bongard

Academic Editor

PLOS ONE

Journal Requirements:

When submitting your revision, we need you to address these additional requirement.

2. PLOS ONE does not copy edit accepted manuscripts (https://journals.plos.org/plosone/s/criteria-for-publication#loc-5). To that effect, please ensure that your submission is free of typos and grammatical errors.

3. We note that Figures 1, 2 & 9 in your submission contain copyrighted images. All PLOS content is published under the Creative Commons Attribution License (CC BY 4.0), which means that the manuscript, images, and Supporting Information files will be freely available online, and any third party is permitted to access, download, copy, distribute, and use these materials in any way, even commercially, with proper attribution. For more information, see our copyright guidelines: http://journals.plos.org/plosone/s/licenses-and-copyright.

 a. You may seek permission from the original copyright holder of Figure(s) [#] to publish the content specifically under the CC BY 4.0 license.

Reviewers' comments:

Reviewer's Responses to Questions

**Comments to the Author**

1. Is the manuscript technically sound, and do the data support the conclusions?

Reviewer #1: Partly

Reviewer #2: Yes

2. Has the statistical analysis been performed appropriately and rigorously? 

Reviewer #1: Yes

Reviewer #2: Yes

3. Have the authors made all data underlying the findings in their manuscript fully available?

Reviewer #1: Yes

Reviewer #2: Yes

4. Is the manuscript presented in an intelligible fashion and written in standard English?

Reviewer #1: Yes

Reviewer #2: Yes

5. Review Comments to the Author

Reviewer #1: The manuscript by Harin and colleagues reported a dyad control task in which two participants controlled the left and right sides of an avatar, respectively. Two participants’ shoulders were connected with braces, so that when one person reached a target and rotated the body, the shoulder at the same side of the other person was pushed forward. The results of the questionnaire showed that ownership rating and agency rating were less negative in the tied condition, compared to the condition when their shoulder was not tied. The three proprioception tests did not show consistent clear changes in the perceived position of the uncontrolled shoulder.

The present study showed that passive synchronized body movement can slightly induce illusionary sense of ownership and agency. The authors suggest that tactile input from the braces contributed to the change in ownership and agency ratings. Did the shoulder position also changed when the brace was moved by the partner? If so, I think it is difficult to say that this is a pure effect of tactile input. In the rubber hand experiment by Kalckert & Ehrsson (2012), participants reported ownership in the passive congruent condition, and the agency rating was also less negative in the passive congruent condition than that in the passive incongruent condition. In addition, Weger et al. (2004) reported that even verbal information regarding a motion is sufficient to cause changes in agency rating. Can the authors please clarify how the present study differs from the previous studies?

Kalckert, A., & Ehrsson, H. H. (2012). Moving a rubber hand that feels like your own: A dissociation of ownership and agency. Frontiers in Human Neuroscience, 6(March), 40. https://doi.org/10.3389/fnhum.2012.00040

Wegner, D. M., Sparrow, B., & Winerman, L. (2004). Vicarious agency: Experiencing control over the movements of others. Journal of Personality and Social Psychology, 86(6), 838–848. https://doi.org/10.1037/0022-3514.86.6.838

Moreover, the authors suggested that the tactile input/passive body movement “improved” the sense of ownership and agency for the uncontrolled body part. I think it is important to carefully describe what the data really showed. The ownership and agency ratings were very negative for the uncontrolled arm. It means that people clearly rejected ownership and agency for the uncontrolled arm. However, when the tactile/motion of the brace was synchronized with the visual input, the rejection was less strong. I don’t think this should be described as that the ownership and agency was “improved”. There wasn’t a sense of ownership or agency for the uncontrolled arm. The changes in rating might have reflected confidence in decision making instead of the feeling of agency and ownership.

The significant proprioception drift (test 1) for the uncontrolled side in the tied condition was interesting. It showed that the synchronized tactile input/passive motion had an effect on the body representation. A similar result was also reported by Kalckert & Ehrsson (2012).

Reviewer #2: The authors examined whether the virtual arm that was not controlled can be embodied using a ‘Joint Avatar’ system that caused connection forces. They found that the sense of agency and ownership towards the virtual arm in Tied condition were significantly higher than Separated condition from the data of the questionnaire for measuring embodiment. From the results, they suggested that the connection force feedback synchronized with the movements of the joint virtual avatar enhances illusory embodiment towards the limb controlled by another. However, three behavioral tests didn’t show any significance between the two conditions. Therefore, I have the following concerns mainly related the point that the effect of the connection forces was only found from the subjective questionnaire.

(1) The authors assume that the connection forces in the ‘Joint Avatar’ setup simulate the forces on the body caused by the movement of independent additional limbs physically connected to the user’s body. However, there are large differences between the push and pull force caused by the connected poles of the experimental setup and the force caused by the movement of the attached limb itself. Although the authors pointed this out and discussed as a possible factor that did not produce results in the behavioral experiments and also discussed as a limitation of the study, the authors should clarify that whether the strong connected force towards the target, which doesn’t occur just from the movement of the attached limb, was necessary for the enhancement of subjective embodiment or not. If the authors claim that the force only by the movement of the attached limb (and not by the poles) causes the enhancement, please explain why it can be generalized from the results with the force by the poles used in this study. Otherwise, please clarify what kind of force is actually needed to increase embodiment for the attached limbs.

(2) Since the differences between Tied and Separated conditions are clearly recognizable to the participants, it is possible that the differences of the sense of ownership and agency between these conditions were caused by higher-order cognitive comparisons and not by the difference of perceptual embodiment itself. If so, the effect of the order would be large. To test this possibility, I recommend separating the data into two different ordered groups and comparing them. If the comparison between the conditions as a between participant design showed difference irrespective of the first or second session, this would be good evidence to support that the significant difference between conditions found in this study were caused by the strength of the sense of ownership and agency themselves and not by the relative difference by some higher-order cognitive judgment. On the other hand, if there was no effect in the first half session and a clear effect in the latter half, the significant difference may possibly be caused by cognitive comparisons, and it would be problematic to claim that the effect of connection force feedback is caused by embodiment itself.

(3) In addition to the connection forces, synchronized proprioceptive feedback of entire upper body movement can be obtained from ‘Joint Avatar’ setup. Therefore, there is a possibility that the synchronized proprioceptive feedback by the movement of other body parts toward targets, despite of no proprioception in the virtual limb itself, contributed to the embodiment. Please consider this possibility and discuss it.

(4) Please explain why the authors conducted two types of tests for body image (and one for body schema) and clarify the purposes and differences among these tests. Also, please explain why test 1 and 2 are related to body image and test 3 is related to body schema. Because the difference between body image and body schema is not simple, please explain in more detail.

In relation to this point, in the sentence in Line 129-131 about body image and body schema, it is unclear what the part “but in the context of limb replacement” means. Please explain in more detail.

(5) In Line 281-285, the authors hypothesized that small movement of the upper body limited the induced sense of ownership. If the authors’ hypothesis is correct, there should be correlation between the amount of movement and the embodiment. If the authors have a log of body positions, please conduct correlation analysis between the amount of movements and the behavioral results (and also the movements and the questionnaire results) to test the hypothesis.

Followings are minor comments.

(1) “Magic tape” should be rephrased as “hook and loop fastener”.

(2) “Head-mount display” in Line 342 should be rephrased as “head mounted display”.

(3) A space should be placed between a number and its unit.

(4) In Line 364, “VR..” (an extra period) should be “VR.”

6. PLOS authors have the option to publish the peer review history of their article (what does this mean?). If published, this will include your full peer review and any attached files.

Reviewer #1: No

Reviewer #2: No

---

## [Author Response · Author response to Decision Letter 0]

26 Jul 2022

Response to reviewer comments

We would like to thank the reviewers for their valuable comments that helped us to improve the quality of the manuscript. We have carefully considered all the comments by the reviewers and provided appropriate responses. Please find herewith, our response to reviewer comments.

Please note that the statements in blue are the comments by the reviewers. Our responses are shown in black.

Reviewer 1

Reviewer #1: The manuscript by Harin and colleagues reported a dyad control task in which two participants controlled the left and right sides of an avatar, respectively. Two participants’ shoulders were connected with braces, so that when one person reached a target and rotated the body, the shoulder at the same side of the other person was pushed forward. The results of the questionnaire showed that ownership rating and agency rating were less negative in the tied condition, compared to the condition when their shoulder was not tied. The three proprioception tests did not show consistent clear changes in the perceived position of the uncontrolled shoulder.

The present study showed that passive synchronized body movement can slightly induce illusionary sense of ownership and agency. The authors suggest that tactile input from the braces contributed to the change in ownership and agency ratings. Did the shoulder position also changed when the brace was moved by the partner? If so, I think it is difficult to say that this is a pure effect of tactile input. 

We agree that the upper body movements must have affected the sense of embodiment towards the non-controlled arm. We supposed that the connection force feedback included the resulting postural changes and did not intend to conclude that the tactile input alone affected the sense of embodiment. However, as you pointed out, the terminology seems inaccurate and misleading. Therefore, we changed the term “connection force feedback” to “connection induced upper body movements” or “passive synchronized upper body movements” (since this type of feedback mainly adjusts the upper body posture of the participant to simulate a feeling of the shoulder and upper body being dragged along with the visually perceived arm movements) throughout the manuscript including the title. Abstract has also been rewritten to avoid misunderstandings.

New title: 

“Effect of connection induced upper body movements on embodiment towards a limb controlled by another during virtual co-embodiment.”

Line:17-30 (Abstract)

“Even if we cannot control them, or when we receive no tactile or proprioceptive feedback from them, limbs attached to our bodies can still provide indirect proprioceptive and haptic stimulations to the body parts they are attached to simply due to the physical connections. In this study we investigated whether such indirect movement and haptic feedbacks from a limb contribute to a feeling of embodiment towards it. To investigate this issue, we developed a 'Joint Avatar' setup in which two individuals were given full control over the limbs in different sides (left and right) of an avatar during a reaching task. The backs of the two individuals were connected with a pair of solid braces through which they could exchange forces and match the upper body postures with one another. Coupled with the first-person view, this simulated an experience of the upper body being synchronously dragged by the partner-controlled virtual arm when it moved. We observed that this passive synchronized upper-body movement significantly reduced the feeling of the partner-controlled limb being owned or controlled by another. In summary, our results suggest that even in total absence of control, connection induced upper body movements synchronized with the visible limb movements can positively affect the sense of embodiment towards partner-controlled or autonomous limbs.”

In the rubber hand experiment by Kalckert & Ehrsson (2012), participants reported ownership in the passive congruent condition, and the agency rating was also less negative in the passive congruent condition than that in the passive incongruent condition. In addition, Weger et al. (2004) reported that even verbal information regarding a motion is sufficient to cause changes in agency rating. Can the authors please clarify how the present study differs from the previous studies?

Kalckert, A., & Ehrsson, H. H. (2012). Moving a rubber hand that feels like your own: A dissociation of ownership and agency. Frontiers in Human Neuroscience, 6(March), 40. https://doi.org/10.3389/fnhum.2012.00040

Wegner, D. M., Sparrow, B., & Winerman, L. (2004). Vicarious agency: Experiencing control over the movements of others. Journal of Personality and Social Psychology, 86(6), 838–848. https://doi.org/10.1037/0022-3514.86.6.838

The force/postural feedback (now renamed as connection induced upper body movement) in our experiment is different from the passive congruent condition in the experiments by Kalckert and Ehrsson (2012, 2014). In their passive congruent condition, they moved the real finger of the participants passively along with the corresponding finger of the rubber hand. If we match that to our experiment, it will be similar to a scenario where the real arm of the participant in the non-controlled side is passively moved (by the partner or the experimenter) along with the virtual limb. Instead, we kept the real arm fixed and passively let the partner move the upper body (the immediate next section of the body on to which the arm is attached) of the participants to investigate if it affected the embodiment towards the virtual arm that moved in VR. We believe this is an unexplored question in previous studies and further explanations have been added to the Introduction and discussion sections about this.

Line: 62-78 (Introduction) 

"But almost all previous embodiment studies have considered scenarios where a real limb is replaced by an artificial limb (like a rubber hand). In these scenarios, tactile feedback[29][31] or even proprioceptive feedback[37][38][39] corresponding to visible touch or movement[38][39] of the artificial limb, can be provided on the real limb it replaces. However, such scenarios are not possible with ‘Independent additional limbs’[40], like in the case of robotics prosthetics or robotics supernumerary limbs, as there is no replacement limb in these scenarios to provide a movement or tactile feedback on. 

Previous works by Kalckert & Ehrsson [41][42] have shown that passive synchronous movement of a participant’s finger along with the corresponding finger of a rubber hand (while the participant observes only the rubber hand) induces an equally strong sense of ownership towards the rubber hand compared to the traditional rubber hand illusion which involves visuo-tactile stimulation with a paint brush. However, would ownership towards a limb be affected if a different body part is passively moved synchronized with an observed finger or a limb? While visual perturbations of independent additional limbs do not lead to tactile or movement feedback (on the corresponding replaced limb), they do lead to forces and hence movements of the body due to the fact that they are physically connected to the user’s body. In this study, we were interested in understanding whether connection induced upper body movements caused by these connection forces can increase the sense of ownership and/or agency towards a partner-controlled limb, and specifically a partner-controlled arm.”

Line: 343-354 (Discussion) 

“Even though the passive movement is already known to induce ownership of artificial limbs[38][41][42], in those previous studies the experimenter has passively moved an artificial finger connected to the real finger of the participant and shown that it induces ownership towards the artificial finger. The sense of agency towards the passively moved finger was less negative when the actual hand and the rubber hand were placed anatomically congruent than when they were incongruent [42]. If we match these to our experiment, it will be similar to a scenario where the real arm of the participant in the non-controlled side is passively moved (by the partner or the experimenter) along with the virtual limb. However, instead here we provide the participants with only the connection induced movements and forces on their shoulder and upper body. Therefore, our study focuses on passive movement of the upper body (or the place where the virtual limb is supposed to join the body) as a result of limb movements by the partner.”

Regarding the work by Wegner et al., (2004), we think it is a highly related study and added a new paragraph to the Discussion section to explain how verbal information is related to feeling of control for arms moved by others and how our study is different. We also mentioned their work in the introduction section as important related work. Thank you for the valuable suggestion. 

Line: 96-103 (Introduction) 

“One study by Wegner et al., (2004)[45] on vicarious agency (that did not use virtual avatars or robotics) has reported proof about experiencing control over the movements of others with a setup where participants watched themselves in a mirror while another person behind them, hidden from view, extended hands forward on each side where participants’ hands would normally appear and performed a series of movements. They have shown that hearing instructions previewing hand movements causes an enhanced feeling of controlling the hands[45]. However, factors other than verbal instructions that may affect control or ownership towards limbs controlled by others remain unexplored.”

Line: 359-370 (Discussion)

“Previous research has shown that verbal information previewing movements of another’s arms placed to seem like a person’s own arms can induce vicarious agency to those arms [45]. While it is possible that the passive upper body movements conveyed cues previewing the movements of the partner’s arm to some degree, the verbal information previewing movement is more cognitive and detailed (for example, “wave hello with your right hand”) compared to the passive upper body movements in our study that informs only the movement timing, direction, and force. Furthermore, the targets reached by the partner were invisible to participants in our experiment making it harder to guess exactly where in space the partner-controlled arm would move. Therefore, influence from previewing information regarding the arm movements were controlled as possible in our experiment. If previewing verbal information about arm movements are also provided along with the passive upper body movements we can expect an even higher score for sense of agency for partner-controlled limbs. This has to be explored in future studies.”

Moreover, the authors suggested that the tactile input/passive body movement “improved” the sense of ownership and agency for the uncontrolled body part. I think it is important to carefully describe what the data really showed. The ownership and agency ratings were very negative for the uncontrolled arm. It means that people clearly rejected ownership and agency for the uncontrolled arm. However, when the tactile/motion of the brace was synchronized with the visual input, the rejection was less strong. I don’t think this should be described as that the ownership and agency was “improved”. There wasn’t a sense of ownership or agency for the uncontrolled arm. The changes in rating might have reflected confidence in decision making instead of the feeling of agency and ownership.

We updated the abstract and conclusion to conclude that passive synchronized upper body movements “reduced the feeling of the partner-controlled limb being owned and controlled by another” instead of mentioning an increase in ownership, agency.

Line: 26-28 (Abstract)

“We observed that this passive synchronized upper-body movement significantly reduced the feeling of the partner-controlled limb being owned or controlled by another.”

Line: 280-285 (Discussion) 

“The results showed that the sense of body ownership towards the non-controlled arm was significantly less negative when the participants were provided connection induced upper body movements synchronized with the movements of the partner-controlled arm during the reaching task compared to when they were not provided any passive movements. Furthermore, the sense of agency towards the non-controlled arm was also less negative when passive synchronized upper body movements were present.”

The significant proprioception drift (test 1) for the uncontrolled side in the tied condition was interesting. It showed that the synchronized tactile input/passive motion had an effect on the body representation. A similar result was also reported by Kalckert & Ehrsson (2012).

Thank you for the suggestion. We newly described that this effect observed in Test 1 is similar to the results reported by Kalckert & Ehrsson (2012) in the discussion.

Line: 301-308 (Discussion) 

"Proprioceptive drifts are known to occur in both active (participant moves the rubber hand) and passive (experimenter simultaneously moves the rubber hand and the participant’s corresponding hand) rubber hand illusions (in the spatially congruent condition of Kalckert & Ehrsson (2012)[42]), suggesting that similar levels of ownership are induced regardless of whether the movements were active or passive when the rubber hand is presented in an anatomically congruent posture [42]. The proprioceptive drift of the non-controlled limb in the Tied condition of Test 1 in our experiment is somewhat consistent with this and interestingly suggests that a certain level of ownership can be induced to limbs even by connection induced synchronous movements of the upper body (instead of the limb itself).”

Reviewer 2

Reviewer #2: The authors examined whether the virtual arm that was not controlled can be embodied using a ‘Joint Avatar’ system that caused connection forces. They found that the sense of agency and ownership towards the virtual arm in Tied condition were significantly higher than Separated condition from the data of the questionnaire for measuring embodiment. From the results, they suggested that the connection force feedback synchronized with the movements of the joint virtual avatar enhances illusory embodiment towards the limb controlled by another. However, three behavioral tests didn’t show any significance between the two conditions. Therefore, I have the following concerns mainly related the point that the effect of the connection forces was only found from the subjective questionnaire.

(1) The authors assume that the connection forces in the ‘Joint Avatar’ setup simulate the forces on the body caused by the movement of independent additional limbs physically connected to the user’s body. However, there are large differences between the push and pull force caused by the connected poles of the experimental setup and the force caused by the movement of the attached limb itself. Although the authors pointed this out and discussed as a possible factor that did not produce results in the behavioral experiments and also discussed as a limitation of the study, the authors should clarify that whether the strong connected force towards the target, which doesn’t occur just from the movement of the attached limb, was necessary for the enhancement of subjective embodiment or not. If the authors claim that the force only by the movement of the attached limb (and not by the poles) causes the enhancement, please explain why it can be generalized from the results with the force by the poles used in this study. Otherwise, please clarify what kind of force is actually needed to increase embodiment for the attached limbs.

We do not claim that the force alone causes the enhancement of embodiment. In addition to the force to the shoulder, the movements of the shoulder (being pushed and pulled at the same time at both shoulders by the partner) also have contributed to the change in embodiment. Considering this, we changed the term “connection force feedback” to “connection induced upper body movements” or “passive synchronized upper body movements”　(which includes the forces as well as movements caused by the back brace setup) throughout the manuscript including the title. Here what we mean by connection induced upper body movement is the movement felt on the upper body as a result of the connection between the partner-controlled limb and the upper body of a participant when the partner-controlled limb moves dragging the body along with it. Our setup simulates this connection induced upper body movements, but it is not ideal compared to the real movements felt on the upper body of an amputee wearing a smart prosthetic arm as we have now mentioned in limitations of the study section. 

New title: 

“Effect of connection induced upper body movements on embodiment towards a limb controlled by another during virtual co-embodiment.

Line:17-30 (Abstract)

“Even if we cannot control them, or when we receive no tactile or proprioceptive feedback from them, limbs attached to our bodies can still provide indirect proprioceptive and haptic stimulations to the body parts they are attached to simply due to the physical connections. In this study we investigated whether such indirect movement and haptic feedbacks from a limb contribute to a feeling of embodiment towards it. To investigate this issue, we developed a 'Joint Avatar' setup in which two individuals were given full control over the limbs in different sides (left and right) of an avatar during a reaching task. The backs of the two individuals were connected with a pair of solid braces through which they could exchange forces and match the upper body postures with one another. Coupled with the first-person view, this simulated an experience of the upper body being synchronously dragged by the partner-controlled virtual arm when it moved. We observed that this passive synchronized upper-body movement significantly reduced the feeling of the partner-controlled limb being owned or controlled by another. In summary, our results suggest that even in total absence of control, connection induced upper body movements synchronized with the visible limb movements can positively affect the sense of embodiment towards partner-controlled or autonomous limbs.”

Line: 274-285 (Discussion) 

“In this study, we investigated whether just the upper-body movements induced by the movements of a virtual arm controlled by another, can affect/enhance embodiment towards it. We developed a joint avatar for this investigation, in which the left and right-limbs were controlled by two different individuals. We compared the senses of body ownership and agency towards the two arms (the controlled arm and the non-controlled arm) of the joint virtual avatar under Tied (with passive synchronized upper body movements) and Separated (without passive synchronized upper body movements) conditions. The results showed that the sense of body ownership towards the non-controlled arm was significantly less negative when the participants were provided connection induced upper body movements synchronized with the movements of the partner-controlled arm during the reaching task compared to when they were not provided any passive movements. Furthermore, the sense of agency towards the non-controlled arm was also less negative when passive synchronized upper body movements were present.”

Line: 433-451 (Limitations of the study) 

“The passive movement by our brace device with support poles was limited to pushing and pulling the non-controlled shoulder which would not exactly be similar to a passive movement felt due to a real prosthetic arm being moved. As we mentioned above, a more sophisticated force feedback device like a robot exoskeleton setup, may further improve the sense of ownership and agency towards the non-controlled arm due to passive movements. 

Furthermore, with the current setup, we cannot dissociate the contributions of postural and haptic feedbacks to the results since postural changes were correlated and in fact caused by the connection forces. Thus, it is an open question whether the haptic feedback alone or the postural feedback of the shoulder alone can affect the sense of embodiment. This should be investigated in a future study.”

(2) Since the differences between Tied and Separated conditions are clearly recognizable to the participants, it is possible that the differences of the sense of ownership and agency between these conditions were caused by higher-order cognitive comparisons and not by the difference of perceptual embodiment itself. If so, the effect of the order would be large. To test this possibility, I recommend separating the data into two different ordered groups and comparing them. If the comparison between the conditions as a between participant design showed difference irrespective of the first or second session, this would be good evidence to support that the significant difference between conditions found in this study were caused by the strength of the sense of ownership and agency themselves and not by the relative difference by some higher-order cognitive judgment. On the other hand, if there was no effect in the first half session and a clear effect in the latter half, the significant difference may possibly be caused by cognitive comparisons, and it would be problematic to claim that the effect of connection force feedback is caused by embodiment itself.

Following your suggestion, we performed two exploratory analyses. One is a three-way ANOVA-ART including the order condition, and the other is two-way ANOVA-ART separately in the first and second sessions. The effect of order was not clear or systematic and did not fully support the higher-order cognitive bias. Thus, we added this carefully in the Discussion section.

Line: 371-422 (Discussion) 

“Since the differences between Tied and Separated conditions are clearly recognizable to the participants, one may argue that the differences of the sense of ownership and agency between these conditions could be caused by higher-order cognitive comparisons and not by the difference of perceptual embodiment itself. If so, the effect of the order would be large. To test this possibility, we performed an exploratory analysis separating the data into two different ordered groups and compared them. If the comparison between the conditions as a between participant design showed a difference irrespective of the first or second session, it supports that the significant difference between conditions found in this study were caused by the strength of the sense of ownership and agency themselves and not by the relative difference by some higher-order cognitive judgment. On the other hand, if there was no effect in the first half session and a clear effect in the latter half, the significant difference may possibly be caused by cognitive comparisons. We performed a mixed-design three-way ANOVA-ART [49] with the condition order (ST: Separated condition first, Tied condition second, TS: Tied condition first, Separated condition second) as a between-subject variable, the condition (Separated condition, Tied condition) as a within-subject variable, and the side (Controlled arm, Non-controlled arm) as a within-subject variable for both the senses of ownership and agency. For the sense of ownership, we found main effects of the condition (F(1,22)=10.015, p=.004, η_p^2=0.315) and the side (F(1,22)=131.71, p<.0001, η_p^2=0.857), and an interaction between the condition and the side (F(1,22)=26.605, p<.0001, η_p^2=0.507), consistent with the original 2-way ANOVA. We did not find a 3-way interaction between condition, side and order (F(1,22)=0.277, p=.604, η_p^2=0.012), suggesting that the condition order did not affect the ownership towards the non-controlled arm in the Tied condition over the Separated condition. We found an interaction between the order and condition (F(1,22)=6.91, p=.015, η_p^2=0.239). However, no simple effects were significant, and the difference between Tied and Separated conditions was larger when the Separated condition was performed first than when the Tied condition was performed first (ST: F(1,35)=2.339, p=.135, η_p^2=0.063, TS: F(1,35)=0.115, p=.736, η_p^2=0.003). For the sense of agency, we found main effects of the condition (F(1,22)=10.537, p=.004, η_p^2=0.324) and the side (F(1,22)=134.37, p<.0001, η_p^2=0.859), and an interaction between the condition and the side (F(1,22)=16.343, p=.0005, η_p^2=0.426), consistent with the original 2-way ANOVA. We did not find a 3-way interaction between condition, side and order (F(1,22)=0.331, p=.571, η_p^2=0.015), suggesting that the condition order would not affect the agency towards the non-controlled arm in the Tied condition over the Separated condition. We did not find any other effects.

To test the effect of order further in detail, we separated the data into the first session and the second session (accordingly the force condition (Tied/separated) is between-subject in this analysis) and performed another exploratory analysis. We performed a mixed-design two-way ANOVA-ART with the condition (Separated condition, Tied condition) as a between-subject factor, and the side (Controlled arm, Non-controlled arm) as a within-subject factor for both senses of ownership and agency. Analyses were performed separately for the first session and the second session. For the sense of ownership, we found a main effect of the side (F(1,22)=99.33, p<.0001, η_p^2=0.819) and an interaction between condition and side (F(1,22)=4.833, p=.0387, η_p^2=0.180) for the first session, similar to the original analysis. However, the simple effect analysis showed that the condition effect on the sense of ownership towards the non-controlled arm in the Tied condition compared to the Separated condition was not significant (F(1,22)=2.490, p=.129, η_p^2=0.102). This may be due to the data division. In the second session, we obtained main effects of condition (F(1,22)=5.350, p=.030, η_p^2=0.196) and side (F(1,22)=89.037, p<.0001, η_p^2=0.802), but the interaction between condition and side was not significant (F(1,22)=3.912, p=.0606, η_p^2=0.151) for the second session. For the sense of agency, we found a main effect of the side (F(1,22)=116.49, p<.0001, η_p^2=0.841), but the interaction between condition and side was not significant (F(1,22)=2.680, p=.1159, η_p^2=0.109) for the first session. In the second session, we obtained main effects of condition (F(1,22)=5.433, p=.0293, η_p^2=0.198) and side (F(1,22)=99.766, p<.0001, η_p^2=0.819), but the interaction between condition and side was not significant (F(1,22)=3.278, p=.0839, η_p^2=0.130) for the second session. To summarize, we cannot conclude that the effect of order never affects the results. However, the results cannot be solely explained by the higher-order cognitive bias because the effect of order was small, and the interaction between the condition and side was significant in the first session, where participants did not know both Separated and Tied conditions.”

(3) In addition to the connection forces, synchronized proprioceptive feedback of entire upper body movement can be obtained from ‘Joint Avatar’ setup. Therefore, there is a possibility that the synchronized proprioceptive feedback by the movement of other body parts toward targets, despite of no proprioception in the virtual limb itself, contributed to the embodiment. Please consider this possibility and discuss it.

We agree with the possibility that the upper body movements and resulting proprioceptive feedbacks may have affected the sense of embodiment towards the non-controlled arm. Thus, we modified the manuscript as explained in the answer to your comment (1). 

With regards to possible proprioceptive information conveyed to other body parts towards the target, a previous work by Wegner et al., has shown that when participants were given previewing instructions about hand movements by another (that was placed like their own), they reported an increased feeling of control to those hands. Therefore, it is possible that proprioceptive cues and drag of upper body towards the target caused an increase in embodiment for the partner’s arm. To minimize this however, we made the targets reached by the partner invisible to the participants of our experiment. However, this is an important issue that should be discussed, and we have now added some discussion regarding this.

We also mentioned the inability of the current study to dissociate the haptic (force) and proprioceptive feedbacks to the limitations section.

Line: 359-370 (Discussion)

“Previous research has shown that verbal information previewing movements of another’s arms placed to seem like a person’s own arms can induce vicarious agency to those arms [45]. While it is possible that the passive upper body movements conveyed cues previewing the movements of the partner’s arm to some degree, the verbal information previewing movement is more cognitive and detailed (for example, “wave hello with your right hand”) compared to the passive upper body movements in our study that informs only the movement timing, direction, and force. Furthermore, the targets reached by the partner were invisible to participants in our experiment making it harder to guess exactly where in space the partner-controlled arm would move. Therefore, influence from previewing information regarding the arm movements were controlled as possible in our experiment. If previewing verbal information about arm movements are also provided along with the passive upper body movements we can expect an even higher score for sense of agency for partner-controlled limbs. This has to be explored in future studies.”

Line: 448-451 (Limitations of the study) 

Furthermore, with the current setup, we cannot dissociate the contributions of postural and haptic feedbacks to the results since postural changes were correlated and in fact caused by the connection forces. Thus, it is an open question whether the haptic feedback alone or the postural feedback of the shoulder alone can affect the sense of embodiment. This should be investigated in a future study.”

(4) Please explain why the authors conducted two types of tests for body image (and one for body schema) and clarify the purposes and differences among these tests. Also, please explain why test 1 and 2 are related to body image and test 3 is related to body schema. Because the difference between body image and body schema is not simple, please explain in more detail.

In relation to this point, in the sentence in Line 129-131 about body image and body schema, it is unclear what the part “but in the context of limb replacement” means. Please explain in more detail.

We added these explanations as follows.

Line: 290-298 (Discussion) 

“Tests 1 and 2 are related to body image since these tests quantified the participant’s perception of their body (specifically location and width of the shoulder in our case) in the visual space. On the other hand, our Test 3 tried to analyze changes in body schema, which refers to the internal (non-visual) representation of one’s body in the proprioceptive and haptic space. In Test 3, participants had their HMDs blacked out (similar to eyes closed) and they used their sensory-motor function of the controlled side arm to point at the shoulder position making test 3 related to body schema. We conducted three types of tests to ensure no changes in body representation goes unobserved due to possible limitations of any one of the tests.”

(5) In Line 281-285, the authors hypothesized that small movement of the upper body limited the induced sense of ownership. If the authors’ hypothesis is correct, there should be correlation between the amount of movement and the embodiment. If the authors have a log of body positions, please conduct correlation analysis between the amount of movements and the behavioral results (and also the movements and the questionnaire results) to test the hypothesis.

We did the correlation analysis you suggested and added the following lines to the discussion.

Line: 320-329 (Discussion) 

“We speculate these limitations in the setup limited the induced sense of ownership in our experiment. To test this speculation, using the body position data of participants recorded during the reaching task, we calculated the distances moved by the real shoulder of participants in the non-controlled side each time the partner performed the reaching in the Tied condition. The average movement of the shoulder (how much the shoulder was pushed by the partner in average) for all participants was calculated and we checked the correlation of this variable with ownership of the non-controlled arm, agency of the non-controlled arm, and drifts in tests 1, 2, and 3 in the tied condition. However, no correlations were seen (Ownership: r(22)=.095, p=.330, Agency: r(22)=.057, p=.396, Test 1: r(22)=-.126, p=.724, Test 2: r(22)=.275, p=.097, Test 3: r(22)=-.268, p=.898). Therefore, this exploratory correlation analysis between the amount of movement and the subjective/behavioral results does not support our speculation.” 

Followings are minor comments.

All of the following have been corrected as you suggested. Thank you.

(1) “Magic tape” should be rephrased as “hook and loop fastener”.

Line: 479

“These back braces were pasted to the bodies of the participants with hook and loop fastener and also tied around the shoulders with elastic bands to fit their bodies and make firm connections.”

(2) “Head-mount display” in Line 342 should be rephrased as “head mounted display”.

Line: 471-472

“Both participants were allowed to freely move their heads and look around in the virtual environment from the first-person perspective of the joint avatar through their Head-Mounted displays (HMDs).”

(3) A space should be placed between a number and its unit.

We kept a space between all numbers and their units throughout the manuscript.

(4) In Line 364, “VR..” (an extra period) should be “VR.”

Line: 481

“This set up induced passively moved the participants to adjust their upper body postures to match the posture of the joint avatar experienced in VR.”

---

## [Decision Letter · Decision Letter 1]

10 Oct 2022

PONE-D-21-38910R1Effect of connection induced upper body movements on embodiment towards a limb controlled by another during virtual co-embodimentPLOS ONE

Dear Dr. Hapuarachchi,

Thank you for submitting your manuscript to PLOS ONE. After careful consideration, we feel that it has merit but does not fully meet PLOS ONE’s publication criteria as it currently stands. Therefore, we invite you to submit a revised version of the manuscript that addresses the points raised during the review process.

We look forward to receiving your revised manuscript.

Kind regards,

Josh Bongard

Academic Editor

PLOS ONE

Reviewers' comments:

Reviewer's Responses to Questions

**Comments to the Author**

1. If the authors have adequately addressed your comments raised in a previous round of review and you feel that this manuscript is now acceptable for publication, you may indicate that here to bypass the “Comments to the Author” section, enter your conflict of interest statement in the “Confidential to Editor” section, and submit your "Accept" recommendation.

Reviewer #1: (No Response)

2. Is the manuscript technically sound, and do the data support the conclusions?

Reviewer #1: Partly

3. Has the statistical analysis been performed appropriately and rigorously? 

Reviewer #1: Yes

4. Have the authors made all data underlying the findings in their manuscript fully available?

Reviewer #1: Yes

5. Is the manuscript presented in an intelligible fashion and written in standard English?

Reviewer #1: Yes

6. Review Comments to the Author

Reviewer #1: Thank you for revising the manuscript. The discussion of the revised version is much improved. I still have some concerns regarding the significance and contribution of this paper. My impression is that the authors somehow came out with this joint avatar paradigm and tested it. This paradigm does not seem to be designed to test any hypotheses or answer any unknown research questions. If the authors wanted to test whether indirect movement and haptic feedback could affect embodiment toward an uncontrolled limb, the pulling force to the partner’s shoulder would be unnecessary. It is also difficult to imagine when such joint embodiment would occur in the real world. I hope the authors could revise the introduction to explain why the research on ‘joint avatar’ and corresponding ‘joint ownership’ would be important.

If we only think from one participant’s perspective, the person could see a virtual limb conducting reaching movements that were controlled by someone else, and received synchronized pushing forces on the corresponding shoulder. This possibly has some effects on the ratings of ownership and agency. However, the question still is, is this novel, and is this important? I’m not fully convinced.

Minor comment:

The authors used he/his to describe the paradigm in the introduction. Although all the participants were male, the paradigm was not designed for male-only. The authors should replace these pronouns.

7. PLOS authors have the option to publish the peer review history of their article (what does this mean?). If published, this will include your full peer review and any attached files.

Reviewer #1: No

---

## [Author Response · Author response to Decision Letter 1]

25 Oct 2022

Reviewer #1: Thank you for revising the manuscript. The discussion of the revised version is much improved. I still have some concerns regarding the significance and contribution of this paper. My impression is that the authors somehow came out with this joint avatar paradigm and tested it. This paradigm does not seem to be designed to test any hypotheses or answer any unknown research questions. 

The key research question motivating this study is the issue of ‘connection’ (and connection related haptic and proprioceptive stimulation). While there is a plethora of studies that combine visual and haptic feedback to test embodiment, in every previous study, the touch is presented visually on the artificial limb (being embodied), and correspondingly real haptic and proprioceptive feedback is presented directly on the innate limb that the artificial limb replaces. The connection related haptic as well as proprioceptive feedback is never on the limb, but rather on the torso to which the limb is connected. Furthermore, connection related haptic feedbacks are self-induced (by self-generated movements) rather than by external objects (or other limbs) touching the limb under observation. And this is the first study to investigate whether feedbacks (corresponding to the connection) that are not really on the limb, can induce a sense of embodiment towards a moving artificial limb. We have further explained this issue in the introduction. 

Line: 68-78 (Introduction)

Previous studies by Kalckert & Ehrsson [41][42] have shown that passive synchronous movement of a participant’s finger along with the corresponding finger of a rubber hand (while the participant observes only the rubber hand) induces an equally strong sense of ownership towards the rubber hand compared to the traditional rubber hand illusion which involves visuo-tactile stimulation with a paint brush. However, if we try to induce embodiment towards an artificial limb or a prosthetic limb of an amputee, the passive synchronous movements of the artificial limb (or the autonomous prosthetic limb) will not provide proprioceptive feedback of the whole limb movements to the amputee. Limb movements will only lead to forces and hence movements of the body since the artificial limb is physically connected to the user’s body. It is unknown whether such connection induced body movements caused by the connection forces can increase the sense of ownership and/or agency towards partner-controlled limbs while the user/amputee observes its movements. 

Then, we added the following lines to the introduction explaining the hypotheses. In the discussion section, we added sentences regarding the hypotheses.

Line: 165-174 (Introduction)

“In summary, the purpose of this experiment was to simulate a feeling similar to the movements felt on the torso of an amputee wearing a prosthetic arm when the prosthetic arm moves by itself and to investigate if such movements influence the senses of ownership and agency felt towards the prosthetic arm. Therefore, we made the following two hypotheses regarding the Tied condition in which such upper body movements were present and the Separated condition in which such upper body moves were absent while the partner-controlled arm moved.

H1: Senses of ownership and agency towards the virtual non-controlled arm would be higher in the Tied condition compared to the Separated condition.

H2: Proprioceptive drifts towards the virtual non-controlled arm would be higher in the Tied condition compared to the Separated condition.”

Line: (Discussion)

Thus, H1 was supported, but the effects of passive synchronized upper body movements or connection force were not large and limited.

Line: (Discussion)

Thus, H2 was partly supported, but we need further discussion and future investigation.

If the authors wanted to test whether indirect movement and haptic feedback could affect embodiment toward an uncontrolled limb, the pulling force to the partner’s shoulder would be unnecessary.

Since our setup provides pulling (as well as pushing) forces to both shoulders of the partner, we are unsure if by “pulling force to the partner’s shoulder”, you meant the pulling force provided to the shoulder of the controlled arm side or the pulling force provided to the non-controlled arm side of the partner. So, we will answer for both cases below.

If you meant the pulling force to the controlled arm side shoulder of the partner,

We think this is an important limitation of our setup that we missed to explain. You are correct that it is unnecessary. But for our setup to work and provide the overall upper body movement, it was necessary. We added explanation to the limitations of the study section. Thank you for pointing this out.

Line: 458-468 (Limitations of the study)

To simulate passive movements on the upper body felt due to a real prosthetic arm being moved, our brace device with support poles provided pushing and pulling forces to the non-controlled shoulder. However, pulling and pushing forces had to be applied to the controlled shoulder as well for this setup to work and provide an overall rotational movement to the upper body since the reaching task involved rotational movements of the upper body to reach targets far away from the avatar. Although these forces to the controlled side makes the simulation deviate from real upper body movements that may be caused by a self-moving/autonomous prosthetic arm, these pulling/pushing forces were necessary for our setup to successfully provide upper body movements to each other in synchrony with the virtual arm controlled by the partner. A more sophisticated force feedback device like a robot exoskeleton setup, may have to be used to simulate a more realistic haptic feedbacks closer to upper body movements caused by the pull of a moving attached autonomous arm and minimize such limitations.

If you meant pulling force to the non-controlled arm shoulder by the partner,

When a real arm moves, we receive both, a pulling force at the shoulder due to the connection of the limb to our body, as well as the resulting series of indirect movements, which includes the forward movement of the upper torso and a twisting of the upper torso, the resulting twist on the hip, and even forces on the legs. Here we aim to reproduce/simulate effects of a connection, and hence all the above effects. Our joint body setup was designed keeping this in mind. Therefore, as we want to reproduce a ‘connection’ and related effects as best as possible, we feel the reproduction of the force at the partner’s shoulder is necessary for this. 

It is also difficult to imagine when such joint embodiment would occur in the real world. I hope the authors could revise the introduction to explain why the research on ‘joint avatar’ and corresponding ‘joint ownership’ would be important.

We revised the introduction with explanations about why research on ‘joint avatar’ and corresponding ‘joint ownership’/embodiment are important.

Line: 153-164 (Introduction)

“Joint/shared/co-embodied avatars and robots can provide means for people with different physical disabilities to combine their strengths with others and work together in one complete avatar/robot to perform complicated tasks effectively while minimizing their physical disabilities at the same time. For example, a person missing a left arm can pair up with a person missing a right arm to co-embody one joint avatar and control different sides together as one whole avatar in VR. However, in such cases, since one half of the body is completely controlled by another, it becomes necessary to figure out ways to increase the sense of embodiment (induce illusory embodiment) to body parts controlled by the partner to minimize discomfort. It is difficult to imagine that two persons are physically connected in the real world. However, the force on the upper body could be conveyed to a person at a distant place with appropriate devices. It is much easier than conveying detailed proprioceptive senses of the whole limb. Thus, our experimental paradigm will contribute to developing joint avatars and robots in the future.”

If we only think from one participant’s perspective, the person could see a virtual limb conducting reaching movements that were controlled by someone else, and received synchronized pushing forces on the corresponding shoulder. This possibly has some effects on the ratings of ownership and agency. However, the question still is, is this novel, and is this important? I’m not fully convinced.

The novelty is in the fact that we test embodiment in scenarios where, instead of the limb being replaced (by the embodied artificial limb), the feedbacks due to the connection are felt on the body to which the limb is attached. We hope it is better conveyed now compared to the previously revised manuscript. Importance of joint avatars and this study is also added as mentioned in the answer for the previous comment. Overall, the Introduction section has been mainly revised taking your comments into account. Thank you.

Minor comment:

The authors used he/his to describe the paradigm in the introduction. Although all the participants were male, the paradigm was not designed for male-only. The authors should replace these pronouns.

Our apologies. This issue is now corrected. We changed term “he” to “participant” in the introduction.

Lines: 85, 86

---

## [Decision Letter · Decision Letter 2]

9 Nov 2022

Effect of connection induced upper body movements on embodiment towards a limb controlled by another during virtual co-embodiment

PONE-D-21-38910R2

Dear Dr. Hapuarachchi,

We’re pleased to inform you that your manuscript has been judged scientifically suitable for publication and will be formally accepted for publication once it meets all outstanding technical requirements.

Kind regards,

Josh Bongard

Academic Editor

PLOS ONE

Additional Editor Comments (optional):

Reviewers' comments:

Reviewer's Responses to Questions

**Comments to the Author**

1. If the authors have adequately addressed your comments raised in a previous round of review and you feel that this manuscript is now acceptable for publication, you may indicate that here to bypass the “Comments to the Author” section, enter your conflict of interest statement in the “Confidential to Editor” section, and submit your "Accept" recommendation.

Reviewer #1: All comments have been addressed

2. Is the manuscript technically sound, and do the data support the conclusions?

Reviewer #1: Yes

3. Has the statistical analysis been performed appropriately and rigorously? 

Reviewer #1: Yes

4. Have the authors made all data underlying the findings in their manuscript fully available?

Reviewer #1: Yes

5. Is the manuscript presented in an intelligible fashion and written in standard English?

Reviewer #1: Yes

6. Review Comments to the Author

Reviewer #1: (No Response)

7. PLOS authors have the option to publish the peer review history of their article (what does this mean?). If published, this will include your full peer review and any attached files.

Reviewer #1: No

---

## [Editor Report · Acceptance letter]

21 Dec 2022

PONE-D-21-38910R2 

Effect of connection induced upper body movements on embodiment towards a limb controlled by another during virtual co-embodiment 

Dear Dr. Hapuarachchi:

I'm pleased to inform you that your manuscript has been deemed suitable for publication in PLOS ONE. Congratulations! Your manuscript is now with our production department. 

Kind regards, 

on behalf of

Dr. Josh Bongard 

Academic Editor

PLOS ONE